# Towards Undistillable Models by Minimizing Conditional Mutual Information

A deep neural network (DNN) is said to be undistillable if, when used as a black-box input-output teacher, it cannot be distilled through knowledge distillation (KD). In this case, the distilled student (referred to as the knockoff student) does not outperform a student trained independently with label smoothing (LS student) in terms of prediction accuracy. To protect intellectual property of DNNs, it is desirable to build undistillable DNNs. To this end, it is first observed that an undistillable DNN may have the trait that each cluster of its output probability distributions in response to all sample instances with the same label should be highly concentrated to the extent that each cluster corresponding to each label should ideally collapse into one probability distribution. Based on this observation and by measuring the concentration of each cluster in terms of conditional mutual information (CMI), a new training method called CMI minimized (CMIM) method is proposed, which trains a DNN by jointly minimizing the conventional cross entropy (CE) loss and the CMI values of all temperature scaled clusters across the entire temperature spectrum. The resulting CMIM model is shown, by extensive experiments, to be undistillable by all tested KD methods existing in the literature. That is, the knockoff students distilled by these KD methods from the CMIM model underperform the respective LS students. In addition, the CMIM model is also shown to performs better than the model trained with the CE loss alone in terms of their own prediction accuracy. The code for the paper is publicly available at `https://github.com/ICLR2024CMIM/CMIM`.

## 1 Introduction

Originally aiming for model compression, knowledge distillation (Buciluǎ et al., 2006; Hinton et al., 2015) (KD) has received significant attention from both academia and industry in recent years due to its remarkable effectiveness. The essence of KD is to transfer the knowledge of a pre-trained large model (teacher) to a smaller model (student). Building on the work of Hinton et al. (2015), numerous follow-up works have endeavored to enhance the performance of KD (Romero et al., 2014; Anil et al., 2018; Park et al., 2019) and to gain deeper insights into why distillation is effective (Phuong & Lampert, 2019; Mobahi et al., 2020; Ye et al., 2024; Allen-Zhu & Li, 2020; Menon et al., 2021; Borup & Andersen, 2021).

In the scenario where the teacher does not want its knowledge to be transferred, however, KD is undesirable and indeed poses a threat to intellectual property (IP) of the teacher (Shokri & Shmatikov, 2015). Developing and training a high-quality large DNN requires significant investments of time, effort, finances, and resources, including intensive data annotation and computational infrastructure. Developers of the large DNN may want to prevent the knowledge of the large DNN from being transferred by their competitors. However, once the large DNN is released as a "black box", anyone can apply a logit-based KD method (or equivalently a distribution-based KD method (Zheng & Yang, 2024)) to distill the DNN as a teacher. The goal is to train a student, referred to as a knockoff student in the context of DNN IP protection, that mimics the teacher's behavior to gain competitive advantages. As such, in this case, it would be desirable for the developers to build the large DNN so that it is undistillable. The question, of course, is how.

Before delving deeper into the above question, let us first clarify what we mean by saying that a DNN is undistillable. At this point, we invoke the concept of distillable DNN introduced recently in Yang & Ye (2024):

**Definition 1.1.** [Distillability of a DNN (Yang & Ye, 2024)] When used as a black-box input-output teacher, a DNN is said to be distillable with respect to a student if there exists a KD method which, when applied to the teacher and student, yields a knockoff student outperforming the student trained alone with label smoothing (LS student) in terms of prediction accuracy.

Therefore, a DNN is undistillable if no knockoff student can outperform the respective LS student regardless of which logit-based KD method is used. Since there are so many logit-based KD methods and so many students, what type of a DNN is undistillable? In comparison with the training of LS student, Definition 1.1 provides some insight into a trait that an undistillable DNN may possess. For each label, consider the cluster of the output probability distributions of the DNN in response to all input sample instances with that label. If each cluster corresponding to each label is highly concentrated to the extent that all probability distributions within the cluster more or less collapse into one probability distribution, then the student training within KD is similar to that of the respective LS student regardless of which logit-based KD method and which student are applied. In this case, one would expect that no knockoff student would perform significantly better than the respective LS student. Therefore, a DNN possessing this trait will likely be undistillable.

Given a DNN, we now measure the concentration of its clusters in terms of conditional mutual information (CMI) (Yang et al., 2023). Specifically, let $X$ denote the random input sample to the DNN, and $Y$ be the ground truth label of $X$. Let $\hat{Y}$ denote the random label predicted by the DNN in response to input $X$. It was shown in Yang et al. (2023) that for each label $y$, the label specific CMI $\mathrm{I}(X; \hat{Y} | Y = y)$ measures the concentration of the cluster corresponding to label $y$, and the CMI $\mathrm{I}(X; \hat{Y} | Y)$ measures the average concentration across all clusters. To build an undistillable DNN, one then is motivated to minimize jointly the conventional cross entropy (CE) loss and the CMI $\mathrm{I}(X; \hat{Y} | Y)$.

In this paper, we will go one step further. In KD (Geoffrey Hinton, 2015), temperature scaling of logits is often applied. It was shown in Zheng & Yang (2024) that logit temperature scaling with temperature $T$ can be equivalently achieved by power transform of the output probability distribution with power $\alpha = 1/T$. Further, it was demonstrated in Ye et al. (2024) that the purpose of temperature scaling or power transform is to enlarge the CMI values of temperature scaled (or power transformed) clusters, and enlarging CMI values in turn improves the performance of distilled students. Since here we want to achieve the opposite, we want to make sure that all CMI values of all power transformed clusters can be made small. To this end, we further extend the label specific CMI $\mathrm{I}(X; \hat{Y} | Y = y)$ and the CMI $\mathrm{I}(X; \hat{Y} | Y)$ to $\mathrm{I}(X; \hat{Y}^\alpha | Y = y)$ and $\mathrm{I}(X; \hat{Y}^{\alpha[Y]} | Y)$, respectively, so that $\mathrm{I}(X; \hat{Y}^\alpha | Y = y)$ measures the concentration of the power transformed cluster corresponding to label $y$ with power $\alpha$, and $\mathrm{I}(X; \hat{Y}^{\alpha[Y]} | Y)$ measures the average concentration across all power transformed clusters with power $\alpha[Y]$, where different clusters may be power transformed with different power $\alpha$.

Based on the above discussion and towards building undistillable DNNs, we then propose a new training method called CMI minimized method, which trains a DNN by jointly minimizing the CE loss and all CMI values of all power transformed clusters, i.e., jointly minimizing the CE loss and $\mathrm{I}(X; \hat{Y}^{\alpha[Y]} | Y)$, $\forall \alpha[Y] > 0$.

The resulting trained DNN is referred to as the CMI minimized (CMIM) DNN. The contributions of the paper are summarized as follows:

• An insight is provided that in order for a DNN to be undistillable, it is desirable for the DNN to possess the trait that each cluster of the DNN's output probability distributions corresponding to each label is highly concentrated to the extent that all probability distributions within the cluster more or less collapse into one probability distribution close to the one-hot probability vector of that label.

• We extend the label specific CMI $\mathrm{I}(X; \hat{Y} | Y = y)$ and the CMI $\mathrm{I}(X; \hat{Y} | Y)$ to $\mathrm{I}(X; \hat{Y}^\alpha | Y = y)$ and $\mathrm{I}(X; \hat{Y}^{\alpha[Y]} | Y)$, respectively, so that $\mathrm{I}(X; \hat{Y}^\alpha | Y = y)$ measures the concentration of the power transformed cluster corresponding to label $y$ with power $\alpha$, and $\mathrm{I}(X; \hat{Y}^{\alpha[Y]} | Y)$ measures the average concentration across all power transformed clusters with power $\alpha[Y]$, where different clusters may be power transformed with different power $\alpha$.

• We develop a novel training method dubbed CMI minimized method to train a DNN by jointly minimizing the CE loss and all CMI values of all power transformed clusters with the resulting trained DNN referred to as the CMIM DNN.

• We show, by extensive experiments over three popular image classification datasets, namely CIFAR-100 (Krizhevsky et al., 2012), TinyImageNet (Le & Yang, 2015) and ImageNet (Deng et al., 2009), that CMIM DNNs have very small CMI values and are indeed undistillable by all tested KD methods existing in the literature. That is, the knockoff students distilled by these KD methods from the CMIM

models underperform the respective LS students. On the other hand, models trained by defense training methods proposed in the literature are all distillable.

• In addition, we show that the CMIM models achieve a higher classification accuracy compared to those trained with the conventional CE loss.

## 2 RELATED WORKS

In this section, we mention some defense methods against the threat posed by knockoff students attempting to steal the IP of pre-trained DNNs via logit-based KD methods. For a thorough review of related works, including detailed discussions about recent logit-based KD methods, please refer to Appendix B. These defense methods can be mainly categorized into two groups: (i) model stealing resistant training methods that specifically train DNNs to reduce the accuracy of knockoff students while maintaining the original classification accuracy of the model (Ma et al., 2021; Wang et al., 2022); and (ii) post-training defense methods that perform minimal perturbations to the pre-trained model's predictions to mislead the knockoff student (Lee et al., 2019; Orekondy et al., 2020; Cheng & Cheng, 2023). Nonetheless, in Section 5, we will show that models trained by all these defense methods are indeed distillable.

## 3 NOTATION AND PRELIMINARIES

### 3.1 NOTATION

The set of real numbers is denoted by $\mathbb{R}$. Vectors are denoted by bold-face letters (e.g., $\boldsymbol{w}$). The $i$-th element of vector $\boldsymbol{w}$ is denoted by $\boldsymbol{w}[i]$. For two vectors $\boldsymbol{u}, \boldsymbol{v} \in \mathbb{R}^C$, the inequality $\boldsymbol{u} \leq \boldsymbol{v}$ implies that $\boldsymbol{u}[i] \leq \boldsymbol{v}[i], \forall i \in [C]$. For a positive integer $K$, let $[K] \triangleq \{1, ...K\}$. Assume that there are $C$ class labels with $[C]$ as the set of class labels. Let $\mathcal{P}([C])$ denote the set of all $C$ dimensional probability distributions. For any two probability distributions $P_1, P_2 \in \mathcal{P}([C])$, the CE and Kullback-Leibler (KL) divergence between $P_1$ and $P_2$ are denoted by $\mathsf{H}(P_1, P_2)$ and $\mathrm{KL}(P_1, P_2)$, respectively. For any $y \in [C]$ and $P \in \mathcal{P}([C])$, write the CE of the one-hot probability distribution corresponding to $y$ and $P$ as $\mathsf{H}(y, P)$.

For any differentiable function $f(\cdot)$, $\nabla_{\boldsymbol{w}} f(\cdot)$ denotes its gradient vector w.r.t. vector $\boldsymbol{w}$.

For any pair of random variables $(X, Y)$, denote its joint probability distribution by $P_{X,Y}(x, y)$ or simply $P(x, y)$ whenever there is no ambiguity, the marginal distribution of $Y$ by $P_Y(y)$, and the conditional distribution of $Y$ given $X = x$ by $P_{Y|X}(\cdot|x)$. The mutual information between two random variables $X$ and $Y$ is denoted by $\mathrm{I}(X, Y)$, and the CMI of $X$ and $Y$ given a third random variable $Z$ is $\mathrm{I}(X, Y|Z)$.

We regard a classification DNN as a mapping from raw data $x \in \mathbb{R}^d$ to a probability distribution $q_x \in \mathcal{P}([C])$. Given a DNN: $x \in \mathbb{R}^d \to q_x$, let $\boldsymbol{\theta}$ denote its weight vector consisting of all its connection weights; whenever there is no ambiguity, we also write $q_x$ as $q_{x,\boldsymbol{\theta}}$.

### 3.2 LABEL SMOOTHING

Label smoothing (LS) (Pereyra et al., 2017) is a regularization technique that prevents peaked output probability distributions, leading to better generalization, by minimizing the objective function:

$$\mathcal{L}^{LS} = (1 - \epsilon)\mathsf{H}(y, q_x) + \epsilon\mathsf{H}(u, q_x), \tag{1}$$

where $u$ is the uniform distribution over $C$ classes, and $\epsilon$ controls the strength of the regularization.

### 3.3 POWER TRANSFORM OF PROBABILITY DISTRIBUTION

In a "black-box" teacher setting, where only the output probability vectors (and not the logits) of the teacher are accessible to the public, applying temperature scaling directly over the logits of the teacher is not feasible in training knockoff students. In this case, KD training can resort to applying "power transformation of probability distribution" directly to the output probability vectors (Zheng &

Yang, 2024). Specifically, given $P \in \mathcal{P}([\mathcal{C}])$, and a non-negative real number $\alpha$, the power transform of $P$ is another probability distribution define as

$$P^{\alpha}[i] = \frac{(P[i])^{\alpha}}{\sum_{j \in [C]} (P[j])^{\alpha}}, \quad \forall i \in [C]. \tag{2}$$

It is not hard to verify that the power transformed probability distribution $P^{\alpha}$ is equal to the softmax of the logits scaled by temperature $T = 1/\alpha$. Therefore, temperature scaling can be equivalently operated directly on the output probability distribution through power transform.

### 3.4 CMI VALUE OF A DNN

As discussed in Yang et al. (2023), for a multi-class classifier $f : x \in \mathbb{R}^d \rightarrow q_x$, let $\hat{Y}$ be the random label predicted by the $f$ with probability $q_X[\hat{Y}]$ in respond to the input $X$. For each cluster corresponding to label $y \in [C]$, we have

$$I(X; \hat{Y}|Y = y) = \sum_x P_{X|Y}(x|y) \left[ \sum_{i=1}^{C} P_{\hat{Y}|XY}(\hat{Y} = i|x, y) \ln \frac{P_{\hat{Y}|XY}(\hat{Y} = i|x, y)}{P_{\hat{Y}|Y}(\hat{Y} = i|Y = y)} \right] \tag{3}$$

$$= \mathbb{E}_{X|Y} \left[ \left( \sum_{i=1}^{C} q_X[i] \ln \frac{q_X[i]}{P_{\hat{Y}|Y}(\hat{Y} = i|Y = y)} \right) |Y = y \right] = \mathbb{E}_{X|Y} \left[ KL(q_X, s_y)|Y = y \right], \tag{4}$$

where $P_{\hat{Y}|XY}(\hat{Y} = i|x, y) = q_x[i]$ follows from the Markov chain $Y \rightarrow X \rightarrow \hat{Y}$, and $s_y = P_{\hat{Y}|Y}(\cdot|y) = \mathbb{E}_{X|Y}[q_X|Y = y]$. $I(X; \hat{Y}|Y = y)$ measures the concentration of the cluster corresponding to label $y \in [C]$. Averaging over all clusters corresponding to all labels $y$, we get

$$I(X; \hat{Y}|Y) = \sum_{y \in [C]} P_Y(y) I(X; \hat{Y}|Y = y) = \mathbb{E}_{XY}[KL(q_X, s_Y)]. \tag{5}$$

$I(X; \hat{Y}|Y)$ measures the average concentration across all clusters.

When the distribution $P_{X,Y}$ is unknown, we can approximate the CMI of $f$ by its empirical value from a data sample (a training dataset or mini-batch thereof) $\mathcal{D} = \{(x_i, y_i)\}_{i=1}^m$. To this end, let $\mathcal{D}_y = \{1 \leq j \leq m : y_j = y\}$. Denote the size of $\mathcal{D}_y$ by $|\mathcal{D}_y|$. The empirical values of each label specific CMI and the CMI can be calculated as follows

$$I^{emp}(X; \hat{Y}|Y = y) = \frac{1}{|\mathcal{D}_y|} \sum_{i \in \mathcal{D}_y} KL(q_{x_i}, s_y^{emp}), \tag{6}$$

$$I^{emp}(X; \hat{Y}|Y) = \frac{1}{m} \sum_{i=1}^{m} KL(q_{x_i}, s_{y_i}^{emp}), \tag{7}$$

$$\text{where} \quad s_y^{emp} = \frac{1}{|\mathcal{D}_y|} \sum_{i \in \mathcal{D}_y} q_{x_i}, \forall y \in [C]. \tag{8}$$

## 4 CMI MINIMIZED METHOD

In this section, we present our CMI minimized method. We begin with extending $I(X; \hat{Y}|Y = y)$ and $I(X; \hat{Y}|Y)$ to the case of power transformed clusters.

### 4.1 INFORMATION QUANTITIES FOR POWER TRANSFORMED CLUSTERS

Consider a classification DNN: $f : x \in \mathbb{R}^d \rightarrow q_x$ which maps input sample instances $x$ with different labels into clusters of probability distributions $q_x$ in the space $\mathcal{P}([C])$, with one cluster per label. For each label $y \in [C]$, apply the power transform with power $\alpha$ to each probability distribution

$q_x$ within the cluster corresponding to the label $y$. Then, we obtain a power transformed cluster. To measure the concentration of the power transformed cluster, we extend $I(X; \hat{Y}|Y = y)$ to the following information quantity

$$I(X; \hat{Y}^{\alpha}|Y = y) = \mathbb{E}_{X|Y}\left[\text{KL}\left(q_X^{\alpha}, s_{y,\alpha}\right)|Y = y\right], \tag{9}$$

where $s_{y,\alpha} = \mathbb{E}_{X|Y}[q_X^{\alpha}|Y = y]$. Note that if we regard $\hat{Y}^{\alpha}$ as the random label predicted by $f$ with probability $q_X^{\alpha}(\hat{Y}^{\alpha})$ in response to the input sample $X$, i.e., given $X$, $\hat{Y}^{\alpha}$ is equal to a label $c$ with probability $q_X^{\alpha}(c)$, $\forall c \in [C]$, then $I(X; \hat{Y}^{\alpha}|Y = y)$ is exactly the CMI between $X$ and $\hat{Y}^{\alpha}$ given $Y = y$. Thus, $I(X; \hat{Y}^{\alpha}|Y = y)$ measures the concentration of the power transformed cluster corresponding to $y$.

Now, we go one step further and allow different clusters to be power transformed with different powers. Suppose that the cluster corresponding to label $y$ is power transformed with power $\alpha[y]$. Let $\hat{Y}^{\alpha[Y]}$ be the random label predicted by $f$ with probability $q_X^{\alpha[Y]}(\hat{Y}^{\alpha[Y]})$ in response to the input sample $X$ given $Y$. That is, given $Y = y$ and $X = x$, $\hat{Y}^{\alpha[Y]}$ is equal to $c$ with probability $q_x^{\alpha[y]}(c)$ for any $c \in [C]$. We can then extend $I(X; \hat{Y}|Y)$ to $I(X; \hat{Y}^{\alpha[Y]}|Y)$

$$I(X; \hat{Y}^{\alpha[Y]}|Y) = \mathbb{E}_{XY}\left[\text{KL}\left(q_X^{\alpha[Y]}, s_{Y,\alpha[Y]}\right)\right], \tag{10}$$

$$= \sum_{y \in [C]} P_Y(y)\left[\mathbb{E}_{X|Y}\left[\text{KL}\left(q_X^{\alpha[y]}, s_{y,\alpha[y]}\right)|Y = y\right]\right] \tag{11}$$

$$= \sum_{y \in [C]} P_Y(y)I(X; \hat{Y}^{\alpha[y]}|Y = y), \tag{12}$$

where for each $y \in [C]$,

$$s_{y,\alpha[y]} = P_{\hat{Y}^{\alpha[Y]}|Y}(\cdot|y) = \sum_x P_{X|Y}(x|y)q_x^{\alpha[y]} = \mathbb{E}_{X|Y}\left[q_X^{\alpha[y]}|Y = y\right]. \tag{13}$$

Note that $I(X; \hat{Y}^{\alpha[Y]}|Y)$ is exactly the CMI between $X$ and $\hat{Y}^{\alpha[Y]}$ given $Y$ and measures the average concentration across all power transformed clusters with power function $\alpha[Y]$. However, $Y$, $X$, and $\hat{Y}^{\alpha[Y]}$ do not form a Markov chain anymore.

When the distribution $P_{X,Y}$ is unknown, we can approximate $I(X; \hat{Y}^{\alpha[Y]}|Y = y)$ and $I(X; \hat{Y}^{\alpha[Y]}|Y)$ by their respective empirical values from a data sample (a training dataset or mini-batch thereof) $\mathcal{D} = \{(x_i, y_i)\}_{i=1}^m$:

$$I^{emp}(X; \hat{Y}^{\alpha[Y]}|Y = y) = \frac{1}{|\mathcal{D}_y|}\sum_{i \in \mathcal{D}_y} \text{KL}(q_{x_i}^{\alpha[y]}, s_{y,\alpha[y]}^{emp}), \tag{14}$$

$$I^{emp}(X; \hat{Y}^{\alpha[Y]}|Y) = \frac{1}{m}\sum_{i=1}^m \text{KL}(q_{x_i}^{\alpha[y_i]}, s_{y_i,\alpha[y_i]}^{emp}), \tag{15}$$

$$\text{where} \quad s_{y,\alpha[y]}^{emp} = \frac{1}{|\mathcal{D}_y|}\sum_{i \in \mathcal{D}_y} q_{x_i}^{\alpha[y]}, \forall y \in [C]. \tag{16}$$

As discussed in Section 1, an undistillable DNN should exhibit the trait that each of these clusters is highly concentrated and ideally collapses into a single probability distribution that closely resembles the one-hot probability vector for that label.

## 4.2 Framework for Minimizing CMI Values of Power Transformed Clusters

Towards building an undistillable DNN, we now train a DNN $f : x \in \mathbb{R}^d \to q_x$ by jointly minimizing the CE loss and all CMI values of all power transformed clusters. Let

$$\boldsymbol{\alpha} = \left[\alpha[1], \alpha[2], \ldots, \alpha[C]\right],$$

and write each $q_x$ as $q_{x,\theta}$. In our CMI minimized method, the objective function we want to minimize is

$$\mathbb{E}_{XY}\Big[\mathsf{H}(Y, q_{X,\theta})\Big] + \lambda \max_{\boldsymbol{\alpha}} \mathrm{I}(X; \hat{Y}^{\boldsymbol{\alpha}[Y]}|Y) \tag{17}$$

where $\lambda > 0$ is a hyper-parameter trading the CE loss with the maximum CMI, and the maximization over $\boldsymbol{\alpha}$ is taken over the region $0 \leq \alpha[i] \leq \beta, 1 \leq i \leq C$. The optimization problem then becomes

$$\min_{\boldsymbol{\theta}} \Big\{ \mathbb{E}_{XY}\Big[\mathsf{H}(Y, q_{X,\theta})\Big] + \lambda \max_{\boldsymbol{\alpha}} \mathrm{I}(X; \hat{Y}^{\boldsymbol{\alpha}[Y]}|Y) \Big\}$$

$$= \min_{\boldsymbol{\theta}} \Big\{ \mathbb{E}_{XY}\Big[\mathsf{H}(Y, q_{X,\theta})\Big] + \lambda \max_{\boldsymbol{\alpha}} \sum_y P_Y[y] \mathrm{I}(X; \hat{Y}^{\alpha[y]}|Y=y) \Big\} \tag{18}$$

$$= \min_{\boldsymbol{\theta}} \Big\{ \mathbb{E}_{XY}\Big[\mathsf{H}(Y, q_{X,\theta})\Big] + \lambda \sum_y P_Y[y] \max_{\alpha[y]} \mathrm{I}(X; \hat{Y}^{\alpha[y]}|Y=y) \Big\}. \tag{19}$$

In order to get a better understanding about the behaviour of the second term in the objective function of equation 18 w.r.t. $\boldsymbol{\alpha}$, we depict in Figure 1 $\mathrm{I}(X; \hat{Y}^{\boldsymbol{\alpha}[Y]}|Y=y)$ vs $\alpha[y]$ for three randomly-selected classes $y$ using a pre-trained ResNet-50 on CIFAR-100. In Figure 1, $\max_\alpha \mathrm{I}(X; \hat{Y}^\alpha|Y=y)$ is achieved at a value of $\alpha$ which is between 0.25 and 0.75. In Theorem E.1 of Appendix E, we further show that for each label $y$, $\mathrm{I}(X; \hat{Y}^\alpha|Y=y)$ as a function of $\alpha$ is continuously differentiable.

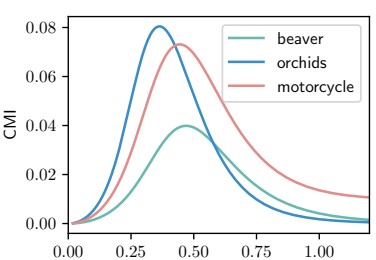

However, finding an algorithmic solution to the min-max problem in equation 18 to equation 19 is challenging. To overcome this difficulty, we next develop a more tractable expression for $\max_\alpha \mathrm{I}(X; \hat{Y}^\alpha|Y=y)$. At this point, we invoke the following theorem, which will be proved in Appendix F.

Figure 1: The class's CMI values for pre-trained ResNet-50 on CIFAR-100 for three randomly selected classes, namely beaver, orchids and motorcycle Vs. the power transform factor $\alpha$.

**Theorem 4.1.** For any label $y$,

$$\max_\alpha \mathrm{I}(X; \hat{Y}^\alpha|Y=y) = \lim_{\omega \to \infty} \frac{1}{\omega} \ln \frac{1}{\beta} \int_0^\beta \exp\{\omega \mathrm{I}(X; \hat{Y}^\alpha|Y=y)\} d\alpha. \tag{20}$$

Therefore, when $\omega$ is large, $\max_\alpha \mathrm{I}(X; \hat{Y}^\alpha|Y=y)$ can be approximated by

$$\max_\alpha \mathrm{I}(X; \hat{Y}^\alpha|Y=y) \approx \frac{1}{\omega} \ln \frac{1}{\beta} \int_0^\beta \exp\{\omega \mathrm{I}(X; \hat{Y}^\alpha|Y=y)\} d\alpha \tag{21}$$

$$\approx \frac{1}{\omega} \ln \left[ \frac{1}{N} \sum_{i=1}^N \exp\{\omega \mathrm{I}(X; \hat{Y}^{\alpha_i}|Y=y)\} \right], \tag{22}$$

where $N$ is relatively large, and $\alpha_i = i\beta/N$.

Now plugging equation 22 into equation 19, we have

$$\min_{\boldsymbol{\theta}} \left\{ \mathbb{E}_{XY}\Big[\mathsf{H}(Y, q_{X,\theta})\Big] + \frac{\lambda}{\omega} \sum_y P_Y[y] \ln \left[ \frac{1}{N} \sum_{i=1}^N \exp\{\omega \mathrm{I}(X; \hat{Y}^{\alpha_i}|Y=y)\} \right] \right\}. \tag{23}$$

Note that the second term in the objective function of equation 23 is not amenable to parallel computation via GPU due to the dependency of KL divergence on $s_{y,\alpha_i}$, the centroid of the power transformed cluster corresponding to $Y=y$ with power $\alpha_i$. To get around this difficulty, we follow the approach in Yang et al. (2023) and introduce dummy distributions $Q_{y,i} \in \mathcal{P}([C])$ for each $(y,i)$ to rewrite $\mathrm{I}(X; \hat{Y}^{\alpha_i}|Y=y)$ as follows

$$\mathrm{I}(X; \hat{Y}^{\alpha_i}|Y=y) = \mathbb{E}_{X|Y}\Big[\mathrm{KL}\Big(q_{X,\theta}^{\alpha_i}, s_{y,\alpha_i}\Big) \mid Y=y\Big]$$

$$= \min_{Q_{y,i}} \mathbb{E}_{X|Y}\Big[\mathrm{KL}\Big(q_{X,\theta}^{\alpha_i}, Q_{y,i}\Big) \mid Y=y\Big], \tag{24}$$

where the minimum in the above is achieved when

$$Q_{y,i} = s_{y,\alpha_i} = \mathbb{E}_{X|Y}\left[q_{X,\boldsymbol{\theta}}^{\alpha_i}|Y = y\right].$$

(25)

Combining equation 24 with equation 23, we are led to solve the double minimization problem

$$= \min_{\boldsymbol{\theta}} \left\{ \mathbb{E}_{XY}\left[\mathsf{H}(Y, q_{X,\boldsymbol{\theta}})\right] + \right.$$

$$\left. \frac{\lambda}{\omega}\sum_y P_Y[y]\ln\left[\frac{1}{N}\sum_{i=1}^N \exp\{\omega\min_{Q_{y,i}}\mathbb{E}_{X|Y}\left[\mathrm{KL}\left(q_{X,\boldsymbol{\theta}}^{\alpha_i}, Q_{y,i}\right)\mid Y = y\right]\}\right]\right\}$$

(26)

$$= \min_{\boldsymbol{\theta}}\min_{\{Q_{y,i}\}_{y\in[C],i\in[N]}} \left\{ \mathbb{E}_{XY}\left[\mathsf{H}(Y, q_{X,\boldsymbol{\theta}})\right] + \right.$$

$$\left. \frac{\lambda}{\omega}\sum_y P_Y[y]\ln\left[\frac{1}{N}\sum_{i=1}^N \exp\{\omega\mathbb{E}_{X|Y}\left[\mathrm{KL}\left(q_{X,\boldsymbol{\theta}}^{\alpha_i}, Q_{y,i}\right)\mid Y = y\right]\}\right]\right\}$$

(27)

When the distribution $P_{X,Y}$ is unknown, it can be approximated by its empirical distribution from a data sample (a training dataset or mini-batch thereof) $\mathcal{D} = \{(x_i, y_i)\}_{i=1}^m$. The objective function in the double minimization equation 27 then becomes

$$J_{\mathcal{D}}(\boldsymbol{\theta}, \{Q_{y,i}\}_{y\in[C],i\in[N]}) = \frac{1}{|\mathcal{D}|}\sum_{(x,y)\in\mathcal{D}}\mathsf{H}(y, q_{x,\boldsymbol{\theta}}) +$$

$$\frac{\lambda}{\omega}\sum_y\frac{|\mathcal{D}_y|}{|\mathcal{D}|}\ln\left[\frac{1}{N}\sum_{i=1}^N\exp\{\frac{\omega}{|\mathcal{D}_y|}\sum_{j\in\mathcal{D}_y}\mathrm{KL}\left(q_{X_j,\boldsymbol{\theta}}^{\alpha_i}, Q_{y,i}\right)\}\right].$$

(28)

### 4.3 Algorithm for Solving the Optimization in equation 27

The double minimization optimization problem in equation 27 naturally lends us an alternating algorithm that optimizes $\boldsymbol{\theta}$ and $\{Q_{y,i}\}_{y\in[C],i\in[N]}$ alternatively to minimize the objective function in equation 27 or Equation (28), given the other is fixed.

Given $\{Q_{y,i}\}_{y\in[C],i\in[N]}$, $\boldsymbol{\theta}$ can be updated using the same first-order optimization method as in conventional deep learning, such as stochastic gradient descent applied over mini-batches.

Following Yang et al. (2023), given $\boldsymbol{\theta}$, for each class $y$, $\{Q_{y,i}\}_{i\in[N]}$ can be updated according to equation 25 in the following manner: (1) we randomly sample a mini-batch of samples $|\mathfrak{B}_y|$ instances from the training set with ground truth label $y$; (2) $\{Q_{y,i}\}_{i\in[N]}$ can be updated as

$$Q_{y,i} = \frac{\sum_{x\in\mathfrak{B}_y}q_{x,\boldsymbol{\theta}}^{\alpha_i}}{|\mathfrak{B}_y|} \quad \forall i \in [N].$$

(29)

The proposed alternating algorithm for optimization problem equation 27 is summarized in Algorithm 1 [1]. To simplify our notation, we use $(\cdot)_b^t$ to indicate parameters at the $b$-th batch updation during the $t$-th alternating iteration of the algorithm. We further write $(\cdot)_B^t$ as $(\cdot)^t$ whenever needed, set $(\cdot)_0^t = (\cdot)^{t-1}$.

## 5 Experiments

In this section, we demonstrate the effectiveness of CMIM by comparing it with several state-of-the-art alternatives. Specifically, we first report the accuracy that a knockoff student can achieve by deploying different logit-based KD (attack) methods in Section 5.1. In all the experiments, when testing the distillibality of the trained DNNs using the benchmark defense methods and CMIM, we

---

[1] If the impact of the random mini-batch sampling and stochastic gradient descent is ignored, the alternating algorithm is guaranteed to converge in theory since given $\boldsymbol{\theta}$, the optimal $\{Q_{y,i}\}_{y\in[C],i\in[N]}$ can be found analytically via equation 29, although it may not converge to a global minimum.

---

**Algorithm 1:** CMIM.

---

**Input:** Training set $\mathcal{T}$, mini-batches $\{\mathcal{B}_b\}_{b\in[B]}$, number of epochs $T$, $\lambda$, $\beta$, $\omega$, $N$

**Initialization:** Initialize $\boldsymbol{\theta}^0$ and $Q^0_{y,i}{}_{y\in[C],i\in[N]}$.

**for** $t = 1$ *to* $T$ **do**

    [Sampling $\alpha_i$] Randomly select N samples $\{\alpha_i\}_{i\in[N]}$ from interval $[0,\beta]$.

    **for** $b = 1$ *to* $B$ **do**

        [Updating $Q_{y,i}$] For each class $y$, construct mini-batch $\{\mathcal{B}_y\}_{y\in[C]}$. Update $Q^t_{y,i}$,

        $\forall y \in [C]$; $\forall i \in [N]$, according to Equation (29).

        [Updating $\boldsymbol{\theta}$] Fix $Q^t_{y,i}{}_{y\in[C],i\in[N]}$. Update $\boldsymbol{\theta}^t_{b-1}$ to $\boldsymbol{\theta}^t_b$ by stochastic gradient descent over the

        objective function 28.

    **end**

**end**

**Output:** Global model $\boldsymbol{\theta}^T$.

---

compare the knockoff student's accuracy (i) when it attempts to steal the IP of protected DNN using logit-based (attack) methods with (ii) when it trains its model using the LS. If the former outperforms the latter, we conclude that the knockoff makes the underlying DNN distillable. Next, in Section 5.2, we report the classification accuracy of the *protected* models trained by the different defense methods. Lastly, in Section 5.3, we visualize the output cluster of models trained by CMIM, CE and NT.

## 5.1 KNOCKOFF STUDENT ACCURACY

• **Datasets:** We conduct extensive experiments on three image classification dataset, namely CIFAR-100 (Krizhevsky et al., 2012) TinyImageNet (Le & Yang, 2015) and ImageNet (Deng et al., 2009). For description of each dataset, please refer to Appendix G.

• **Models:** To show the effectiveness of CMIM, we use different model architectural families for teacher and knockoff student models. To this end, we pick models from VGG family (Simonyan & Zisserman, 2015), ResNet family (He et al., 2016) (shortened as RN), ShuffleNetV2 (Ma et al., 2018), shortened as SNV2, and Mobilenetv2 (Sandler et al., 2018) shortened as MNV2. Particularly, we have conducted experiments on the following (teacher-student) pairs for each dataset: (i) for CIFAR-100, we use four pairs {(VGG16-VGG11), (VGG16-SNV2), (RN50-VGG11), (RN50-RN18)}; (ii) for TinyImageNet, we use two pairs {(RN34-RN18), (RN50-SNV2)}; and for ImageNet we use two pairs {(RN34-RN18),(RN34-MNV2)}.

• **Defense benchmark methods:** For comprehensive comparisons, we benchmark CMIM with seven recently published defense methods: MAD Orekondy et al. (2020), APGP (Cheng & Cheng, 2023), RSP (Lee et al., 2019), ST (Ma et al., 2022), NT (Ma et al., 2021), SNT (Wang et al., 2022), and LS[2] (Müller et al., 2019).

• **Logit-based KD (attack) methods:** We use three logit-based KD methods that are primarily designed for when the teacher-student models are in cooperating mode, namely KD (Hinton et al., 2015), DKD (Zhao et al., 2022), DIST (Huang et al., 2022a); and four logit-based KD attacks methods that a knockoff student can deploy to make the protected DNNs possibly distillable, namely MKD (Yang & Ye, 2024), HTC (Jandial et al., 2022), AVG (Keser & Toreyin, 2023), Knockoff (Orekondy et al., 2019). We report all the training setups, including all the hyper-parameters used for both defense and attack methods in Appendix H.1.

• **Results:** The accuracy that a knockoff student can attain using the above *(defense-attack)* combinations are listed in Table 1 (for the accuracy variances, please refer to Appendix J), where we use the notations K-student to denote knockoff student. The numbers in the column titled "Best" represent the maximum value for each respective row, indicating the highest accuracy that the knockoff student can achieve using the distillation methods.

---

[2]Although LS is not a defense method per se, it is observed that the models trained by LS reduce the knockoff student's accuracy. We discuss the rationale behind this in Appendix C.

Table 1: Top-1 accuracy (%) of the knockoff student on CIFAR-100, TinyImageNet and ImageNet dataset (the results for CIFAR-100 and TinyImageNet are averaged over 3 runs). Green upward arrows (↑) and red downward arrows (↓) indicate whether the knockoff student was able to render the underlying DNN distillable.

| Defense | Model | K-student | LS | KD | MKD | DKD | DIST | HTC | AVG | Knockoff | Best |
|---|---|---|---|---|---|---|---|---|---|---|---|
| **CIFAR-100** | | | | | | | | | | | |
| MAD | VGG16 | VGG11 | 71.94 | 68.55↓ | 72.08↑ | 53.32↓ | 69.21↓ | 71.19↓ | 70.03↓ | 61.44↓ | 72.08↑ |
| | | SNV2 | 72.65 | 72.50↓ | 72.46↓ | 7.64↓ | 69.91↓ | 71.37↓ | 72.86↑ | 70.87↓ | 72.86↑ |
| | RN50 | VGG11 | 71.94 | 72.00↑ | 72.04↑ | 54.29↓ | 71.57↓ | 70.76↓ | 70.73↓ | 61.73↓ | 72.04↑ |
| | | RN18 | 78.76 | 77.76↓ | 78.79↑ | 43.73↓ | 73.76↓ | 77.89↓ | 78.61↓ | 73.92↓ | 78.79↑ |
| APGP | VGG16 | VGG11 | 71.94 | 71.92↑ | 72.27↑ | 27.24↓ | 69.25↓ | 70.08↓ | 72.01↑ | 45.98↓ | 72.27↑ |
| | | SNV2 | 72.65 | 73.10↑ | 73.75↑ | 12.52↓ | 71.04↓ | 71.66↓ | 73.20↑ | 9.48↓ | 73.75↑ |
| | RN50 | VGG11 | 71.94 | 71.91↓ | 72.11↑ | 9.74↓ | 69.48↓ | 71.36↓ | 71.92↓ | 34.71↓ | 72.11↑ |
| | | RN18 | 78.76 | 78.04↓ | 79.06↑ | 62.71↓ | 77.32↓ | 77.82↓ | 77.90↓ | 2.57↓ | 79.06↑ |
| RSP | VGG16 | VGG11 | 71.94 | 71.42↓ | 72.04↑ | 70.22↓ | 70.80↓ | 70.40↓ | 71.56↓ | 31.04↓ | 72.04↑ |
| | | SNV2 | 72.65 | 73.55↑ | 72.95↑ | 67.45↓ | 72.19↓ | 71.46↓ | 72.27↓ | 26.09↓ | 73.55↑ |
| | RN50 | VGG11 | 71.94 | 71.97↑ | 72.01↑ | 69.53↓ | 72.18↑ | 70.87↓ | 70.85↓ | 46.68↓ | 72.18↑ |
| | | RN18 | 78.76 | 77.78↓ | 77.79↓ | 77.01↓ | 78.88↑ | 78.00↓ | 78.13↓ | 55.86↓ | 78.88↑ |
| NT | VGG16 | VGG11 | 71.94 | 71.40↓ | 73.44↑ | 71.47↓ | 71.33↓ | 70.77↓ | 71.58↓ | 63.56↓ | 73.44↑ |
| | | SNV2 | 72.65 | 72.44↓ | 72.70↑ | 6.24↓ | 72.04↓ | 70.75↓ | 72.83↑ | 6.32↓ | 72.83↑ |
| | RN50 | VGG11 | 71.94 | 72.01↑ | 72.03↑ | 71.55↓ | 71.88↓ | 70.16↓ | 71.94↑ | 62.94↓ | 72.03↑ |
| | | RN18 | 78.76 | 78.41↓ | 78.92↑ | 79.26↑ | 78.99↑ | 77.94↓ | 78.33↓ | 68.96↓ | 79.26↑ |
| SNT | VGG16 | VGG11 | 71.94 | 72.06↑ | 72.28↑ | 4.92↓ | 71.98↑ | 70.60↓ | 71.63↓ | 64.08↓ | 72.06↑ |
| | | SNV2 | 72.65 | 72.94↑ | 73.17↑ | 72.78↑ | 72.22↓ | 71.22↓ | 72.74↑ | 6.22↓ | 73.17↑ |
| | RN50 | VGG11 | 71.94 | 72.02↑ | 72.12↑ | 72.32↑ | 71.70↓ | 70.66↓ | 71.65↓ | 62.94↓ | 72.32↑ |
| | | RN18 | 78.76 | 78.25↓ | 78.48↓ | 78.82↑ | 78.14↓ | 78.45↓ | 78.38↓ | 67.71↓ | 78.82↑ |
| ST | VGG16 | VGG11 | 71.94 | 72.09↑ | 72.01↑ | 71.63↓ | 71.93↓ | 71.16↓ | 71.63↓ | 63.32↓ | 72.09↑ |
| | | SNV2 | 72.65 | 72.64↓ | 72.67↑ | 70.53↓ | 72.24↓ | 71.32↓ | 72.42↓ | 69.46↓ | 72.67↑ |
| | RN50 | VGG11 | 71.94 | 72.00↑ | 72.13↑ | 71.62↓ | 71.76↓ | 70.54↓ | 71.73↓ | 65.43↓ | 72.13↑ |
| | | RN18 | 78.76 | 78.96↑ | 79.02↑ | 78.35↓ | 78.31↓ | 78.36↓ | 78.81↑ | 72.87↓ | 79.02↑ |
| LS | VGG16 | VGG11 | 71.94 | 71.90↓ | 72.00↑ | 71.57↓ | 70.89↓ | 70.66↓ | 71.76↓ | 63.49↓ | 72.00↑ |
| | | SNV2 | 72.65 | 72.87↑ | 73.52↑ | 70.01↓ | 71.49↓ | 71.70↓ | 73.01↑ | 65.20↓ | 73.52↑ |
| | RN50 | VGG11 | 71.94 | 71.82↓ | 71.99↑ | 71.95↓ | 70.77↓ | 70.86↓ | 71.88↓ | 62.29↓ | 71.99↑ |
| | | RN18 | 78.76 | 77.72↓ | 77.82↓ | 79.37↑ | 78.33↓ | 78.31↓ | 77.91↓ | 63.36↓ | 79.37↑ |
| CMIM | VGG16 | VGG11 | 71.94 | 71.87↓ | 71.64↓ | 71.56↓ | 70.34↓ | 71.71↓ | 71.42↓ | 66.89↓ | 71.87↓ |
| | | SNV2 | 72.65 | 72.53↓ | 71.44↓ | 72.46↓ | 71.45↓ | 71.59↓ | 71.94↓ | 64.45↓ | 72.53↓ |
| | RN50 | VGG11 | 71.94 | 71.54↓ | 71.34↓ | 71.77↓ | 71.86↓ | 69.32↓ | 71.70↓ | 60.58↓ | 71.86↓ |
| | | RN18 | 78.76 | 78.21↓ | 78.16↓ | 78.13↓ | 77.56↓ | 77.23↓ | 78.64↓ | 65.88↓ | 78.64↓ |
| **TinyImageNet** | | | | | | | | | | | |
| RSP | RN34 | RN18 | 63.56 | 63.54↓ | 64.32↑ | 64.01↑ | 63.27↓ | 63.54↓ | 62.15↓ | 55.43↓ | 64.32↑ |
| | RN50 | SNV2 | 60.61 | 60.18↓ | 60.76↑ | 56.26↓ | 56.43↓ | 60.96↑ | 60.15↓ | 54.01↓ | 60.96↑ |
| ST | RN34 | RN18 | 63.56 | 63.96↑ | 64.12↑ | 63.25↓ | 63.51↓ | 63.49↓ | 63.84↑ | 57.42↓ | 64.12↑ |
| | RN50 | SNV2 | 60.61 | 61.23↑ | 61.36↑ | 60.43↓ | 60.32↓ | 60.22↓ | 61.13↑ | 55.84↓ | 61.36↑ |
| NT | RN34 | RN18 | 63.56 | 63.27↓ | 64.49↑ | 64.67↑ | 63.43↓ | 63.50↓ | 64.43↑ | 53.11↓ | 64.67↑ |
| | RN50 | SNV2 | 60.61 | 59.57↓ | 61.55↑ | 31.55↓ | 60.03↓ | 60.98↑ | 60.31↓ | 50.94↓ | 61.55↑ |
| LS | RN34 | RN18 | 63.56 | 63.74↑ | 64.01↑ | 64.23↑ | 63.51↓ | 64.20↑ | 63.04↓ | 57.43↓ | 64.23↑ |
| | RN50 | SNV2 | 60.61 | 60.32↓ | 60.93↑ | 60.74↑ | 60.11↓ | 60.46↓ | 60.14↓ | 52.96↓ | 60.93↑ |
| CMIM | RN34 | RN18 | 63.53 | 62.89↓ | 63.15↓ | 62.94↓ | 63.28↓ | 61.57↓ | 62.96↓ | 56.13↓ | 63.28↓ |
| | RN50 | SNV2 | 60.61 | 57.57↓ | 59.32↓ | 60.58↓ | 59.41↓ | 59.33↓ | 60.42↓ | 56.91↓ | 60.58↓ |
| **ImageNet** | | | | | | | | | | | |
| ST | RN34 | RN18 | 70.89 | 70.74↓ | 71.02↑ | 70.02↓ | 69.94↓ | 70.91↑ | 71.00↑ | 63.24↓ | 71.02↑ |
| | | MNV2 | 70.93 | 71.03↑ | 71.25↑ | 69.32↓ | 70.53↓ | 70.69↓ | 71.06↑ | 54.53↓ | 71.25↑ |
| CMIM | RN34 | RN18 | 70.89 | 70.44↓ | 70.69↓ | 69.97↓ | 70.59↓ | 70.63↓ | 70.53↓ | 59.34↓ | 70.69↓ |
| | | MNV2 | 70.93 | 70.21↓ | 70.72↓ | 69.97↓ | 70.44↓ | 70.86↓ | 70.20↓ | 55.24↓ | 70.86↓ |

As observed in Table 1, whether the architectures of teacher-student pairs are the same or different, unlike the DNNs trained by other defense methods, the DNNs trained by CMIM cannot be made distillable using different distillation methods.

## 5.2 ACCURACY OF PROTECTED MODELS

In this section, we report the top-1 accuracy of the *protected* models in Table 1 trained using the benchmark defense methods with those trained by CMIM. The results are summarized in Tables 2 and 3. As observed, the models trained by CMIM have the highest classification accuracy compared to the benchmark methods. This is because for the models trained by CMIM, the clusters corresponding to the output probability of the DNNs are very concentrated, facilitating easier classification of samples from different classes.

Table 2: Top-1 accuracy (%) of models trained by defense methods on CIFAR-100 and TinyImageNet. The best and second best results are **bolded** and underlined, respectively.

| CIFAR100 | | | | | | | | | | TinyImageNet | | | | | | |
|---|---|---|---|---|---|---|---|---|---|---|---|---|---|---|---|---|
| Model | CE | MAD | APGP | RSP | ST | NT | SNT | LS | CMIM | Model | CE | RSP | ST | NT | LS | CMIM |
| VGG16 | 73.75 | 73.75 | 73.84 | 73.71 | 73.75 | 73.75 | 72.59 | **73.90** | 73.84 | RN34 | 65.39 | 65.21 | 65.39 | 65.23 | 65.45 | **65.99** |
| RN50 | 77.81 | 77.81 | 77.56 | 77.63 | 77.81 | 77.31 | 77.77 | 78.45 | **78.72** | RN50 | 66.14 | 65.91 | 66.13 | 66.06 | 66.09 | **66.93** |

Table 3: Top-1 accuracy (%) of models trained by defense methods on ImageNet.

| ImageNet | | | |
|---|---|---|---|
| Model | CE | ST | CMIM |
| RN34 | 73.31 | 73.30 | **73.69** |

(a) CE                    (b) Nasty teacher                    (c) CMIM

Figure 2: Visualization of three projected probability clusters for ResNet-50 trained on CIFAR-100 using (a) CE, (b) NT, and (c) CMIM.

The results in Table 2 motivate us to test the top-1 accuracy of additional models trained by CMIM and compare them with those trained by CE loss (see Appendix I).

## 5.3 VISUALIZING THE OUTPUT CLUSTERS

In this subsection, we aim to visualize the output clusters for the models trained by CE, NT and CMIM. to this end, we follow the visualization approach introduced in Yang et al. (2023), and pick three labels randomly from CIFAR-100 dateset. For each probability distribution in the three clusters corresponding to these picked labels, consider only the probabilities of these three labels, normalize them so that they become a three-dimensional probability vector, and further project the resulting probability vector into the two-dimensional simplex. Then the three clusters corresponding to three selected labels are projected into and can be viewed in the two dimensional simplex (Yang et al., 2023). Such simplexes are depicted in Figure 2 for ResNet-50 trained using CE loss, nasty teacher and CMIM framework, where for better visualization we used the same power transform $\alpha = 4$ to depict all the simplexes. As observed, for the model trained by CMIM, the clusters are highly concentrated in the corner of the simplex (one-hot vectors). Thus, a knockoff student cannot outperform LS regularization when distilling a CMIM-trained model.

## 6 CONCLUSION

In this paper, from an information-theoretic perspective, we proposed a defence method against the threat posed by knockoff students attempting to steal the IP of pre-trained DNNs via logit-based KD methods. In particular, we proposed to minimize the CMI of the protected DNN across different power transform hyper-parameter values $\alpha$, while minimizing the conventional CE loss simultaneously. We referred to model trained by these framework as CMIM models. By conducting a series of experiments, we showed that, unlike the prior defense methods proposed in the literature, a knockoff student cannot render CMIM models distillable. In addition, we showed that the models trained by CMIM achieve higher classification accuracy compared to those trained by CE loss.

Despite these promising results, our work has certain limitations. First, the evaluation of CMIM models is primarily empirical, as providing a formal theoretical proof of undistillability remains an open challenge. Second, our approach introduces additional computational overhead compared to the conventional training using CE loss.

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

# A   APPENDIX

# B   A THOROUGH STUDY OF RELATED WORKS

## B.1   LOGIT-BASED KD METHODS

Knowledge Distillation (KD) has become a cornerstone technique for compressing large teacher models into smaller student models. This section reviews key logit-based KD methods and explores their advancements.

Hinton et al. (2015) introduced the foundational concept of KD, using KL divergence to match the student's softmax outputs to the teacher's. This work laid the groundwork for logit-based methods, as the softmax output directly relates to the logits. Later, Moldovan et al. (2019) proposed Path-KD, a method that utilizes the paths leading to the correct class in both teacher and student models for distillation. While not directly logit-based, it demonstrates the effectiveness of aligning decision-making processes. Guo et al. (2021) proposed Logit-Like Distillation, addressing the capacity gap by matching the ranking of logits instead of their exact values [4]. This approach allows the student to learn the essential ordering of classes even with limited capacity. An et al. (2021) proposed relation knowledge distillation (RKD), focusing on aligning relationships between class logits rather than individual values. This approach improves the student's ability to generalize to unseen data. Zhao et al. (2022) introduced decoupled knowledge distillation (DKD), where it decouples the classical KD loss into two parts: target class knowledge distillation and non-target class knowledge distillation. Huang et al. (2022b) proposed DIST where they designed a KD method to distill better from a stronger teacher; indeed they claim that preserving the relations between the predictions of teacher and student would suffice for an effective KD. Borup & Andersen (2021) provided theoretical arguments for the importance of weighting the teacher outputs with the ground-truth targets when performing self-distillation with kernel ridge regressions along with a closed form solution for the optimal weighting parameter.

## B.2   DEFENSE METHODS AGAINST LOGIT-BASED KD

As also discussed in Section 2, the defense methods against the threat posed by knockoff students attempting to steal the IP of pre-trained DNNs via logit-based KD methods can be categorized into two approaches. Here, we elaborate on these two approaches.

**(I) Model stealing resistant training:** In this approach, DNNs are trained to reduce the accuracy of knockoff students while maintaining the original classification accuracy of the model. In particular, Ma et al. (2021) proposed a training algorithm named self-undermining KD to create nasty teachers (NT) that prevent knowledge leakage and unauthorized model stealing through KD, without compromising model accuracy. The nasty teacher is trained by minimizing the following objective function:

$$\mathcal{L}^{NT} = \mathsf{H}(y, q_x) - \epsilon \, \mathrm{KL}(\tilde{q}_x, q_x), \tag{30}$$

where $\tilde{q}_x$ is a output of a pre-trained standard model.

Subsequently, Wang et al. (2022) proposed semantic nasty teachers (SNT) which improve the model stealing resistance of NT by disentangling semantic relationships in the output logits during teacher model training, which is crucial for successful KD.

**(II) post-training defence methods:** The aim of these approaches is to deceive the knockoff by imposing minimal perturbations to the model's predictions. Lee et al. (2019) tested a variety of possible perturbation forms, and found that the reverse sigmoid perturbation (RSP) to be the most effective one. Orekondy et al. (2020) introduced maximizing angular deviation (MAD), a technique that perturbs the output probabilities, leading to an adversarial gradient signal that deviates significantly from the original gradient of the knockoff. To this end, they applied a randomly initialized model as the surrogate for the potential knockoff. More recently, Cheng & Cheng (2023) proposed a plug-and-play generative perturbation model, dubbed as accuracy preserving generative perturbation (APGP), which can effectively defend KD-based model cloning, while preserve the model utility.

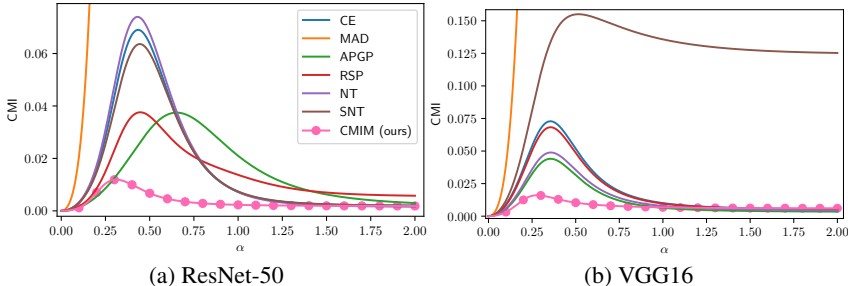

(a) ResNet-50           (b) VGG16

Figure 3: The CMI values for the models trained by different KD-resistance defence methods Vs. power transform value $\alpha$ for (a) ResNet-50, and (b) VGG16 trained on CIFAR-100 dataset.

### B.3 ATTACK METHODS USING LOGIT-BASED KD

Jandial et al. (2022) sought to circumvent the defense of nasty teachers and steal (or extract) its information. Specifically, they analyzed nasty teacher from two different angles and subsequently leverage them carefully to develop simple yet efficient methodologies, named as HTC and SCM, which enhance learning from nasty teacher.

In AVG (Keser & Toreyin, 2023), the authors noted that undistillable teachers exhibit multiple peaks in their softmax response, which are transferred to the student models. These peaks are considered to be the primary factor that misleads the student models. To mitigate the influence of the multiple peaks in the softmax response of teachers, they proposed transferring the mean of features with the same labels as the soft labels.

Orekondy et al. (2019) introduced a technique called "Knockoff Nets" that allows an attacker to steal the functionality of black-box models. Remarkably, the attacker only needs to interact with the model by feeding it input data and observing the resulting predictions. By training a new model ("knockoff") on these input-prediction pairs, the attacker can create a copycat model that performs similarly to the original black box.

## C WHY LS REDUCE THE KNOCKOFF STUDENT'S ACCURACY?

Original aiming to prevent overfitting and improve generalization, label smoothing was observed by Müller et al. (2019) to reduce the accuracy of the knockoff student. The researchers found that "label smoothing encourages examples to lie in tight, equally separated clusters". Consequently, label smoothing reduces the contextual information in the teacher model's output (Yang et al., 2024).

## D WHY PRIOR DEFENCE METHODS CAN BE MADE DISTILLABLE?

In this section, we answer to this question that why the DNNs trained using all prior KD-resistance defense methods could be made distillable (as shown in Section 5). Indeed, the reason lies in the fact that by appropriately adjusting the power transform value $\alpha$, the DNNs trained using these defense methods can potentially achieve a high CMI value compared to our proposed method CMIM (see Figure 3). Thus, using this specific $\alpha$ value during distillation, a logit-based KD method can render these DNNs distillable effectively.

## E POWER TRANSFORMATION OF MUTUAL INFORMATION

The following theorem implies that for each label $y$, $I(X; \hat{Y}^\alpha | Y = y)$ as a function of $\alpha$ is continuously differentiable.

**Theorem E.1.** Let $(X, Z)$ be a pair of random variables, where $Z$ is discrete, and $X$ can be either discrete or continuous. Let $P_{Z|X}[\cdot|x]$ denote the conditional probability distribution of $Z$ given $X = x$. Additionally, let $P_{Z|X}^\alpha[\cdot|x]$ denote the power transformed version of $P_{Z|X}[\cdot|x]$ with power $\alpha$, $Z^\alpha$ denote the random variable the conditional distribution of which given $X = x$ is $P_{Z|X}^\alpha[\cdot|x]$,

and $q^\alpha$ denote the probability distribution of $Z^\alpha$. Then, the following holds

$$\frac{\partial \, \mathrm{I}(X; Z^\alpha)}{\partial \, \alpha} = \frac{1}{\alpha} \sum_x P_X[x] \Big\{ \big( m_2(P_{Z|X}^\alpha[\cdot|x]) - m_1^2(P_{Z|X}^\alpha[\cdot|x]) \big) - \mathrm{Cov}(P_{Z|X}^\alpha[\cdot|x], q^\alpha) \Big\}, \quad (31)$$

where for probability vectors $P$ and $Q$,

$$m_1(P) \triangleq \sum_j P[j]\big( -\ln(P[j]) \big) = \mathsf{H}(P), \quad \text{(Shannon entropy)} \tag{32a}$$

$$m_2(P) \triangleq \sum_j P[j]\big( -\ln(P[j]) \big)^2, \quad \text{(Second moment)} \tag{32b}$$

$$\mathrm{Cov}(P,Q) \triangleq \sum_j P[j]\Big( -\ln(P[j]) - m_1(P) \Big)\Big( -\ln(Q[j]) - \sum_i P[i](-\ln(Q[i])) \Big). \tag{32c}$$

*Proof.* To simplify our notation, we denote the conditional distributions $P_{Z|X}[\cdot|x]$ and $P_{Z|X}^\alpha[j|x]$ by $p_x$ and $p_x^\alpha$, respectively. Decompose $\mathrm{I}(X; Z^\alpha)$ as follows

$$\mathrm{I}(X; Z^\alpha) = H(Z^\alpha) - H(Z^\alpha|X)$$
$$= \mathrm{H}(q^\alpha) - \sum_x P[x]\mathrm{H}(p_x^\alpha) \tag{33}$$

where for any random variables $U$ and $V$, $H(V)$ and $H(V|U)$ are the entropy of $V$ and the conditional entropy of $V$ given $U$, respectively, and $H(p)$ denotes the entropy of the probability distribution $p$. Then the partial derivative of $\mathrm{I}(X; Z^\alpha)$ w.r.t. $\alpha$ is equal to

$$\frac{\partial \mathrm{I}(X; Z^\alpha)}{\partial \alpha} = \frac{\partial \mathrm{H}(q^\alpha)}{\partial \alpha} - \sum_x P[x]\frac{\partial \mathrm{H}(p_x^\alpha)}{\partial \alpha}. \tag{34}$$

To continue, we first compute the partial derivative in the second term of the RHS of equation 34

$$\frac{\partial \mathrm{H}(p_x^\alpha)}{\partial \alpha} = \frac{-\partial \sum_j p_x^\alpha[j]\ln(p_x^\alpha[j])}{\partial \alpha} = -\sum_j \Big( \ln(p_x^\alpha[j]) + 1 \Big)\frac{\partial p_x^\alpha[j]}{\partial \alpha}$$

$$= -\sum_j \Big( \ln(p_x^\alpha[j]) + 1 \Big)$$

$$\times \frac{(p_x[j])^\alpha \ln(p_x[j])\Big( \sum_i (p_x[i])^\alpha \Big) - (p_x[j])^\alpha \Big( \sum_i (p_x[i])^\alpha \ln p_x[i] \Big)}{\Big( \sum_i (p_x[i])^\alpha \Big)^2}$$

$$= -\sum_j \Big( \ln(p_x^\alpha[j]) + 1 \Big)\Big( p_x^\alpha[j]\big( \ln(p_x[j]) - \sum_i p_x^\alpha[i]\ln(p_x[i]) \big) \Big)$$

$$= \frac{-1}{\alpha}\sum_j \Big( \ln(p_x^\alpha[j]) + 1 \Big)p_x^\alpha[j]\Big( \ln(p_x[j])^\alpha - \sum_i p_x^\alpha[i]\ln(p_x[i])^\alpha \Big) \tag{35}$$

$$= \frac{-1}{\alpha}\sum_j \Big( \ln(p_x^\alpha[j]) + 1 \Big)p_x^\alpha[j]\Big( \ln(p_x^\alpha[j]) - \sum_i p_x^\alpha[i]\ln(p_x^\alpha[i]) \Big)$$

$$= \frac{-1}{\alpha}\Big( \sum_j p_x^\alpha[j]\big( \ln(p_x^\alpha[j]) \big)^2 - \Big( \sum_j p_x^\alpha[j]\ln(p_x^\alpha[j]) \Big)\Big( \sum_i p_x^\alpha[i]\ln(p_x^\alpha[i]) \Big) \Big)$$

$$= \frac{-1}{\alpha}\Big( m_2(p_x^\alpha) - m_1^2(p_x^\alpha) \Big) \tag{36}$$

Note that

$$q^\alpha = \sum_x P[x]p_x^\alpha.$$

Then we have

$$\frac{\partial H(q^\alpha)}{\partial \alpha} = \frac{-\partial \sum_j q^\alpha[j] \ln(q^\alpha[j])}{\partial \alpha} = -\sum_j \left( \ln(q^\alpha[j]) + 1 \right) \frac{\partial q^\alpha[j]}{\partial \alpha}$$

$$= -\sum_j \left( \ln(q^\alpha[j]) + 1 \right) \sum_x P[x] \frac{\partial p_x^\alpha[j]}{\partial \alpha}$$

$$= \frac{-1}{\alpha} \sum_j \left( \ln(q^\alpha[j]) + 1 \right) \sum_x P[x] p_x^\alpha[j] \left( \ln(p_x^\alpha[j]) + m_1(p_x^\alpha) \right) \quad (37)$$

$$= \frac{-1}{\alpha} \sum_j \left( \ln(q^\alpha[j]) \right) \sum_x P[x] p_x^\alpha[j] \left( \ln(p_x^\alpha[j]) + m_1(p_x^\alpha) \right)$$

$$= \frac{-1}{\alpha} \sum_x P[x] \sum_j \left( \ln(q^\alpha[j]) \right) p_x^\alpha[j] \left( \ln(p_x^\alpha[j]) + m_1(p_x^\alpha) \right)$$

$$= \frac{-1}{\alpha} \sum_x P[x] \left( \sum_j p_x^\alpha[j] \ln(p_x^\alpha[j]) \left( \ln(q^\alpha[j]) \right) \right.$$

$$\left. - m_1(p_x^\alpha) \sum_j p_x^\alpha[j] \left( - \ln(q^\alpha[j]) \right) \right)$$

$$= \frac{-1}{\alpha} \sum_x P[x] \, \mathrm{Cov} \left( p_x^\alpha, q^\alpha \right) \quad (38)$$

where equation 37 is due to equation 35.

From Equations (36) and (38), Theorem E.1 follows. □

## F    PROOF OF THEOREM 4.1

Theorem 4.1 follows from Theorem E.1 and the following lemma.

**Lemma F.1.** Let $g(t)$ be a continuously differentiable function over $[0, \beta]$. Then the following holds:

$$\max_t g(t) = \lim_{\omega \to \infty} \frac{1}{\omega} \ln \frac{1}{\beta} \int_0^\beta \exp \left\{ \omega g(t) \right\} dt. \quad (39)$$

*Proof.* Let $t^*$ be an optimal point at which

$$g(t^*) = \max_t g(t).$$

For any $\epsilon > 0$, let $\mathcal{N}(t^*, \epsilon)$ denote a closed interval containing $t^*$ with length $\epsilon$. It is easy to verify that

$$\frac{1}{\omega} \ln \frac{1}{\beta} \int_0^\beta \exp \left\{ \omega g(t) \right\} dt \le g(t^*)$$

which implies that

$$\limsup_{\omega \to \infty} \frac{1}{\omega} \ln \frac{1}{\beta} \int_0^\beta \exp \left\{ \omega g(t) \right\} dt \le g(t^*). \quad (40)$$

On the other hand,

$$\frac{1}{\omega} \ln \frac{1}{\beta} \int_0^\beta \exp \left\{ \omega g(t) \right\} dt \ge \frac{1}{\omega} \ln \frac{1}{\beta} \int_{\mathcal{N}(t^*, \epsilon)} \exp \left\{ \omega g(t) \right\} dt$$

$$\ge \frac{1}{\omega} \ln \frac{\epsilon}{\beta} \exp \left\{ \omega \min_{t \in \mathcal{N}(t^*, \epsilon)} g(t) \right\}$$

$$= \min_{t \in \mathcal{N}(t^*, \epsilon)} g(t) + \frac{1}{\omega} \ln \frac{\epsilon}{\beta}. \quad (41)$$

Letting $\omega \to \infty$ in equation 41 yields

$$\liminf_{\omega \to \infty} \frac{1}{\omega} \ln \frac{1}{\beta} \int_0^\beta \exp\{\omega g(t)\} dt \geq \min_{t \in \mathcal{N}(t^*,\epsilon)} g(t). \tag{42}$$

Note that equation 42 is valid for any $\epsilon > 0$. Letting $\epsilon \to 0$ in equation 42, we have

$$\liminf_{\omega \to \infty} \frac{1}{\omega} \ln \frac{1}{\beta} \int_0^\beta \exp\{\omega g(t)\} dt \geq g(t^*). \tag{43}$$

Then equation 39 follows from equation 40 and equation 43. This completes the proof of Lemma F.1.
$\square$

## G  DATASETS DESCRIPTION

- CIFAR-100 (Krizhevsky et al., 2012) dataset contains 50K training and 10K test color images, each with size $32 \times 32$, categorized into 100 classes.
- TinyImageNet (Le & Yang, 2015) contains 120K color images across 200 classes, each with a resolution of $64 \times 64$ pixels. For each class, there are 500 training images, 50 validation images and 50 test images.
- ImageNet (Deng et al., 2009) is a large-scale dataset used in visual recognition tasks, containing around 1.2 million training and 50K validation images.

## H  EXPERIMENTS SETUP

All experiments detailed in this paper were conducted using a publicly available national high-performance computer. For each experiment, we utilized 16 CPU cores, 64 GB of memory, and one NVIDIA V100 GPU. The software environment comprised Python 3.10, PyTorch 1.13, and CUDA 11.

For all experiments, including defenses and attacks, the SGD optimizer (Robbins & Monro, 1951; LeCun et al., 2002) with a learning rate of 0.1 is used unless otherwise specified.

For the CIFAR-100 and TinyImageNet datasets, we train the model for 200 epochs, decaying the learning rate by 0.1 at epochs 60, 120, 160.

For ImageNet, we follow the standard PyTorch practice [3].

The batch size is 128 for both CIFAR-100 and TinyImageNet, and 256 for ImageNet.

To get the accuracy that a knockoff student can achieve using label smoothing, we have tested a wide spectrum of label smoothing factor $\epsilon = \{0.01, 0.05, 0.1, 0.2, 0.3, 0.4, 0.5, 0.6, 0.7, 0.8, 0.9\}$, and selected the one that resulted in the highest classification accuracy.

In the CMIC method, we set $T = 20$ and tested $\lambda = 0.1, 0.25, 0.5, 1$, selecting the value that minimized the CMI value while maintaining or improving classification accuracy.

### H.1  DEFENSE SETUP

We used the following parameters and settings for the defense models used in Section 5.

#### H.1.1  DEFENSE SETUP ON CIFAR-100 AND TINYIMAGENET

1. **MAD**: We employ a randomly initialized VGG-8 as adversary's architecture, and following the implementation of MAD-argmax.
2. **APGP**: We apply a 3 layer MLP with residual connection as the generative model and set $\lambda = 0.1$ for all experiments.
3. **RSP**: We use $\alpha = 1$ and $\beta = 20$ for all the experiments.

---

[3] https://github.com/pytorch/vision/tree/main/references/classification

4. **NT**: To ensure a acceptable accuracy sacrifice, we test three different $\epsilon$ values and select the largest one that results in an accuracy drop of less than $0.5\%$. Specifically, we use $\epsilon = 0.01$ for ResNet-50 and $\epsilon = 0.005$ for VGG-16 on the CIFAR-100 dataset, while for TinyImageNet, we use $\epsilon = 0.001$ for both ResNet-34 and ResNet-50.

5. **SNT**: We use the pretrained word2vect model namely "fasttext-wiki-news-subwords-300" provided by Gensim (Rehurek & Sojka, 2011), and set $\lambda = 0.2$ for all experiments.

6. **ST**: We use the teacher model trained by CE as the underlying model, and use the sparse ratio of $10\%$ as suggested in their paper for all experiments.

7. **LS**: We apply label smoothing factor 0f $0.05$ for all experiments.

8. **ISTM**: We set the binary search parameters to $T_b = 20$ and $\alpha_{\max} = 2000$. We use $\lambda = 0.2$ for ResNet-50 and $\lambda = 0.5$ for VGG-16 on the CIFAR-100 dataset, while for TinyImageNet, we use $\lambda = 0.1$ for ResNet-34 and $\lambda = 0.5$ ResNet-50.

### H.1.2 DEFENSE SETUP ON IMAGENET

1. **ST**: We use the teacher model trained by CE as the underlying model, and use the sparse ratio of $10\%$ as suggested in their paper for all experiments.

2. **ISTM**: We set the binary search parameters to $T_b = 20$ and $\alpha_{\max} = 2000$. We use $\lambda = 0.2$ for all the experiments.

## H.2 ATTACK SETUP

### H.2.1 ATTACK SETUP ON CIFAR-100 AND TINYIMAGENET

We use power transform parameter $\alpha = 0.25$ (or equivalently $T = 4$) for all experiments unless otherwise specified.

1. **KD**: We set the CE-KL trade-off coefficient to $\lambda = 0.9$.

2. **MKD**: We use the intrinsic dimension of 3 for CIFAR-100, and 5 for TinyImageNet. We employed the Adam optimizer (Kingma & Ba, 2014) with learning rate $10^{-3}$ for the trainable Markov transform.

3. **DKD**: We test alpha, beta pairs of $\{1, 4\}$ and $\{2, 8\}$, and report the one with best accuracy.

4. **DIST**: We $use \beta = 1.0, \gamma = 1.0, \tau = 1.0$ for all experiments.

5. **HTC**: We use $\alpha = 0.05(T = 20)$, $\lambda = 0.01$ for all experiments.

6. **AVG**: $\lambda = 0.9$.

7. **Knockoff**: We follow the implementation of the original paper.

### H.2.2 ATTACK SETUP ON IMAGENET

We use $\alpha = 1$ $(T = 1)$ for all experiments unless otherwise specified.

1. **KD**: $\lambda = 0.9$.

2. **MKD**: We use the intrinsic dimension of 16. We employed the Adam optimizer (Kingma & Ba, 2014) with learning rate $10^{-3}$ for the trainable Markov transform.

3. **DKD**: We test alpha, beta pairs of $\{1, 4\}$ and $\{2, 8\}$, and report the one with best accuracy.

## I ACCURACY OF PROTECTED MODELS

In this section, we report the top-1 accuracy of some additional models that are trained using CMIM and compare them with those trained by CE method. To this end, we use 10 well-known models for CIFAR-100 dataset namely ResNet (RN)-$\{18, 34, 50, 101, 152\}$, SqueezeNet (SQN), ResNext (RNXT) 50, MobileNet (MN), Xception (XCP), DenseNet (DN) 121; and 2 models namely RN-$\{34, 50\}$ for TinyImageNet and ImageNet. We follow the same training recipe as the one in Section 5.1. The results for CIFAR-100 and (Tiny-)ImageNet are listed in Table 4 and Table 5, respectively. As seen, the top-1 accuracy for all models trained by CMIC is consistently higher than those trained by CE counterpart, with the gain up to $1.15\%$.

Table 4: Top-1 accuracy (%) of models trained by CE and CMIM methods on CIFAR-100.

| CIFAR-100 | | | | | |
|---|---|---|---|---|---|
| Model | CE | CMIC | Model | CE | CMIC |
| RN18 | 76.05 | 77.20 | SQN | 69.32 | 70.64 |
| RN34 | 77.20 | 77.54 | RNXT50 | 78.71 | 79.12 |
| RN50 | 77.81 | 77.93 | MN | 67.26 | 67.51 |
| RN101 | 79.07 | 79.12 | XCP | 77.37 | 77.64 |
| RN152 | 79.21 | 79.43 | DN121 | 79.16 | 79.33 |

Table 5: Top-1 accuracy (%) of models trained by CE and CMIM methods on TinyImageNet and ImageNet.

| TinyImageNet | | | ImageNet | | |
|---|---|---|---|---|---|
| Model | CE | CMIC | Model | CE | CMIC |
| RN34 | 65.39 | 65.99 | RN34 | 73.31 | 73.69 |
| RN50 | 66.14 | 66.93 | RN50 | 76.15 | 76.40 |

## J Variance of Table 1

Table 6: Top-1 accuracy (%) and variance of the knockoff student on CIFAR-100 and TinyImageNet dataset (averaged over 3 runs)

| | | | | | | CIFAR-100 | | | |
|---|---|---|---|---|---|---|---|---|---|
| Defense | Model | K-student | LS | KD | MKD | DKD | DIST | HTC | AVG | Knockoff |
| MAD | VGG16 | VGG11 | 71.94±0.09 | 68.55±0.20 | 72.08±0.29 | 53.32±0.38 | 69.21±0.09 | 71.19±0.03 | 70.03±0.07 | 61.44±0.06 |
| | | SNV2 | 72.65±0.18 | 72.50±0.11 | 72.46±0.13 | 7.64 ± 0.70 | 69.91±0.18 | 71.37±0.24 | 72.86±0.18 | 70.87±0.26 |
| | RN50 | VGG11 | 71.94±0.09 | 72.00±0.21 | 72.04±0.13 | 54.29±0.52 | 71.57±0.31 | 70.76±0.20 | 70.73±0.22 | 61.73±0.23 |
| | | RN18 | 78.76±0.08 | 77.76±0.23 | 78.79±0.23 | 43.73±0.55 | 73.76±0.10 | 77.89±0.25 | 78.61±0.15 | 73.92±0.19 |
| APGP | VGG16 | VGG11 | 71.94±0.09 | 71.92±0.19 | 72.27±0.21 | 27.24±0.54 | 69.25±0.14 | 70.08±0.23 | 72.01±0.20 | 45.98±0.45 |
| | | SNV2 | 72.65±0.18 | 73.10±0.27 | 73.75±0.23 | 12.52±0.30 | 71.04±0.31 | 71.66±0.13 | 73.20±0.27 | 9.48 ± 0.73 |
| | RN50 | VGG11 | 71.94±0.09 | 71.91 ± 0.17 | 72.11 ± 0.23 | 9.74 ± 0.86 | 69.48 ± 0.11 | 71.38 ± 0.25 | 71.92 ± 0.14 | 34.71 ± 0.30 |
| | | RN18 | 78.76 ± 0.08 | 78.04 ± 0.21 | 79.06 ± 0.14 | 62.71 ± 0.29 | 77.32 ± 0.13 | 77.82 ± 0.13 | 77.90 ± 0.15 | 2.57 ± 0.95 |
| RSP | VGG16 | VGG11 | 71.94 ± 0.09 | 71.42 ± 0.24 | 72.04 ± 0.13 | 70.22 ± 0.19 | 70.80 ± 0.17 | 70.40 ± 0.17 | 71.56 ± 0.06 | 31.04 ± 0.62 |
| | | SNV2 | 72.65 ± 0.18 | 73.55 ± 0.34 | 72.95 ± 0.36 | 67.45 ± 0.20 | 72.19 ± 0.44 | 71.46 ± 0.41 | 72.27 ± 0.27 | 26.09 ± 0.40 |
| | RN50 | VGG11 | 71.94 ± 0.09 | 71.97 ± 0.21 | 72.01 ± 0.20 | 69.53 ± 0.12 | 72.18 ± 0.21 | 70.87 ± 0.14 | 70.85 ± 0.17 | 46.68 ± 0.60 |
| | | RN18 | 78.76 ± 0.08 | 77.78 ± 0.09 | 77.79 ± 0.16 | 77.01 ± 0.09 | 78.88 ± 0.21 | 78.00 ± 0.26 | 78.13 ± 0.12 | 55.86 ± 0.18 |
| NT | VGG16 | VGG11 | 71.94 ± 0.09 | 71.40 ± 0.34 | 73.44 ± 0.16 | 71.47 ± 0.14 | 71.33 ± 0.18 | 70.77 ± 0.23 | 71.58 ± 0.09 | 63.56 ± 0.16 |
| | | SNV2 | 72.65 ± 0.18 | 72.44 ± 0.43 | 72.70 ± 0.35 | 6.24 ± 0.51 | 72.04 ± 0.19 | 70.75 ± 0.13 | 72.83 ± 0.20 | 6.32 ± 0.26 |
| | RN50 | VGG11 | 71.94 ± 0.09 | 72.01 ± 0.25 | 72.03 ± 0.19 | 71.55 ± 0.36 | 71.88 ± 0.31 | 70.16 ± 0.29 | 71.94 ± 0.18 | 62.94 ± 0.24 |
| | | RN18 | 78.76 ± 0.08 | 78.41 ± 0.25 | 78.92 ± 0.14 | 79.26 ± 0.29 | 78.99 ± 0.14 | 77.94 ± 0.22 | 78.33 ± 0.05 | 68.96 ± 0.18 |
| SNT | VGG16 | VGG11 | 71.94 ± 0.09 | 72.06 ± 0.22 | 72.28 ± 0.12 | 4.92 ± 0.22 | 71.98 ± 0.18 | 70.60 ± 0.13 | 71.63 ± 0.10 | 64.08 ± 0.19 |
| | | SNV2 | 72.65 ± 0.18 | 72.94 ± 0.41 | 73.17 ± 0.13 | 72.78 ± 0.20 | 72.22 ± 0.24 | 71.22 ± 0.18 | 72.74 ± 0.20 | 6.22 ± 0.59 |
| | RN50 | VGG11 | 71.94 ± 0.09 | 72.02 ± 0.19 | 72.12 ± 0.39 | 72.32 ± 0.33 | 71.70 ± 0.39 | 70.66 ± 0.17 | 71.65 ± 0.20 | 62.94 ± 0.29 |
| | | RN18 | 78.76 ± 0.08 | 78.25 ± 0.05 | 78.48 ± 0.24 | 78.82 ± 0.30 | 78.14 ± 0.28 | 78.45 ± 0.15 | 78.38 ± 0.13 | 67.71 ± 0.20 |
| ST | VGG16 | VGG11 | 71.94 ± 0.09 | 72.09 ± 0.23 | 72.01 ± 0.14 | 71.63 ± 0.16 | 71.93 ± 0.12 | 71.16 ± 0.27 | 71.63 ± 0.18 | 63.32 ± 0.14 |
| | | SNV2 | 72.65 ± 0.18 | 72.64 ± 0.15 | 72.67 ± 0.21 | 70.53 ± 0.48 | 72.24 ± 0.39 | 71.32 ± 0.38 | 72.42 ± 0.12 | 69.46 ± 0.28 |
| | RN50 | VGG11 | 71.94 ± 0.09 | 72.00 ± 0.19 | 72.13 ± 0.13 | 71.62 ± 0.24 | 71.76 ± 0.29 | 70.54 ± 0.33 | 71.73 ± 0.11 | 65.43 ± 0.24 |
| | | RN18 | 78.76 ± 0.08 | 78.96 ± 0.26 | 79.02 ± 0.06 | 78.35 ± 0.09 | 78.31 ± 0.14 | 78.36 ± 0.25 | 78.81 ± 0.14 | 72.87 ± 0.08 |
| LS | VGG16 | VGG11 | 71.94 ± 0.09 | 71.90 ± 0.18 | 72.00 ± 0.06 | 71.57 ± 0.26 | 70.89 ± 0.12 | 70.66 ± 0.17 | 71.76 ± 0.10 | 63.49 ± 0.21 |
| | | SNV2 | 72.65 ± 0.18 | 72.87 ± 0.28 | 73.52 ± 0.25 | 70.01 ± 0.32 | 71.49 ± 0.38 | 71.70 ± 0.42 | 73.01 ± 0.27 | 65.20 ± 0.14 |
| | RN50 | VGG11 | 71.94 ± 0.09 | 71.82 ± 0.28 | 71.99 ± 0.16 | 71.95 ± 0.33 | 70.77 ± 0.39 | 70.86 ± 0.24 | 71.88 ± 0.16 | 62.29 ± 0.10 |
| | | RN18 | 78.76 ± 0.08 | 77.72 ± 0.30 | 77.82 ± 0.12 | 79.37 ± 0.19 | 78.33 ± 0.06 | 78.31 ± 0.21 | 77.91 ± 0.07 | 63.36 ± 0.17 |
| CMIM | VGG16 | VGG11 | 71.94 ± 0.09 | 71.87 ± 0.24 | 71.64 ± 0.07 | 71.56 ± 0.03 | 70.34 ± 0.09 | 71.71 ± 0.14 | 71.42 ± 0.05 | 66.89 ± 0.11 |
| | | SNV2 | 72.65 ± 0.18 | 72.53 ± 0.21 | 71.44 ± 0.16 | 72.46 ± 0.20 | 71.45 ± 0.31 | 71.59 ± 0.24 | 71.94 ± 0.20 | 64.45 ± 0.24 |
| | RN50 | VGG11 | 71.94 ± 0.09 | 71.54 ± 0.30 | 71.34 ± 0.16 | 71.77 ± 0.06 | 71.86 ± 0.28 | 69.32 ± 0.09 | 71.70 ± 0.22 | 60.58 ± 0.17 |
| | | RN18 | 78.76 ± 0.08 | 78.21 ± 0.13 | 78.16 ± 0.09 | 78.13 ± 0.06 | 77.56 ± 0.06 | 77.23 ± 0.09 | 78.64 ± 0.06 | 65.88 ± 0.09 |
| | | | | | | TinyImageNet | | | | |
| RSP | RN34 | RN18 | 63.56 ± 0.06 | 63.54 ± 0.09 | 64.32 ± 0.07 | 64.01 ± 0.07 | 63.27 ± 0.16 | 63.54 ± 0.07 | 62.15 ± 0.14 | 55.43 ± 0.12 |
| | RN50 | SNV2 | 60.61 ± 0.15 | 60.18 ± 0.26 | 60.76 ± 0.16 | 56.26 ± 0.16 | 56.43 ± 0.11 | 60.96 ± 0.22 | 60.15 ± 0.20 | 54.01 ± 0.22 |
| ST | RN34 | RN18 | 63.56 ± 0.06 | 63.96 ± 0.13 | 64.12 ± 0.07 | 63.25 ± 0.10 | 63.51 ± 0.14 | 63.49 ± 0.19 | 63.84 ± 0.08 | 57.42 ± 0.08 |
| | RN50 | SNV2 | 60.61 ± 0.15 | 61.23 ± 0.24 | 61.36 ± 0.14 | 60.43 ± 0.14 | 60.32 ± 0.24 | 60.22 ± 0.17 | 61.13 ± 0.13 | 55.84 ± 0.11 |
| NT | RN34 | RN18 | 63.56 ± 0.06 | 63.27 ± 0.14 | 64.49 ± 0.17 | 64.67 ± 0.16 | 63.43 ± 0.20 | 63.50 ± 0.10 | 64.43 ± 0.11 | 53.11 ± 0.06 |
| | RN50 | SNV2 | 60.61 ± 0.15 | 59.57 ± 0.22 | 61.55 ± 0.12 | 31.55 ± 0.28 | 60.03 ± 0.23 | 60.98 ± 0.27 | 60.31 ± 0.17 | 50.94 ± 0.15 |
| LS | RN34 | RN18 | 63.56 ± 0.06 | 63.74 ± 0.08 | 64.01 ± 0.14 | 64.23 ± 0.11 | 63.51 ± 0.07 | 64.20 ± 0.16 | 63.04 ± 0.13 | 57.43 ± 0.10 |
| | RN50 | SNV2 | 60.61 ± 0.15 | 60.32 ± 0.24 | 60.93 ± 0.15 | 60.74 ± 0.26 | 60.11 ± 0.28 | 60.46 ± 0.14 | 60.14 ± 0.21 | 52.96 ± 0.23 |
| CMIM | RN34 | RN18 | 63.53 ± 0.06 | 62.89 ± 0.03 | 63.15 ± 0.08 | 62.94 ± 0.03 | 63.28 ± 0.05 | 61.57 ± 0.06 | 62.96 ± 0.06 | 56.13 ± 0.04 |
| | RN50 | SNV2 | 60.61 ± 0.15 | 57.57 ± 0.20 | 59.32 ± 0.17 | 60.58 ± 0.12 | 59.41 ± 0.09 | 59.33 ± 0.10 | 60.42 ± 0.04 | 56.91 ± 0.05 |

## K Computational Overhead

In this section we compare the computational overhead of our method with the CE counterpart on CIFAR-100 dataset.

Table 7: Training times of CMIM and CE for ResNet-10 and VGG-16 on the CIFAR-100 dataset.

| | CE | CMIM |
|---|---|---|
| RN50 | 4 hours 43 minutes | 5 hours 13 minutes |
| VGG16 | 2 hours 57 minutes | 3 hours 25 minutes |

Note that the training time for CMIM is slightly higher than that of conventional CE method. This is primarily due to the additional inference samples required to estimate the CMI. In addition, note that the number of samples $N$ does not have any effects on the training time CMIM; this is because the power transform applied to the teacher's output probabilities, and when calculating the gradients during the backpropagation, different values of $\alpha$ does not change the gradients.

## L Ablation on hyperparameters

In this section, we conduct ablation study on the hyperparameters of range of $\beta$ and number of power samples $\alpha$.

For all experiments of ablation study, we use the VGG-16 - SNV2 as the teacher student pair on the Cifar-100 dataset.

## L.1 RANGE OF $\beta$

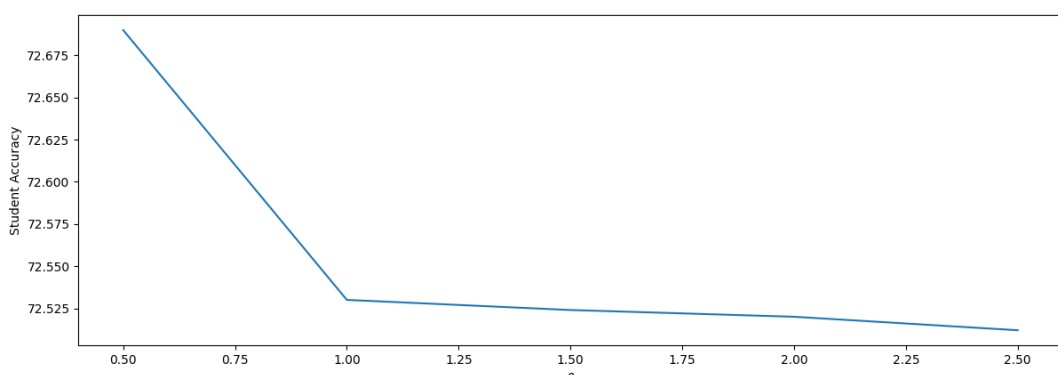

Figure 4: The student accuracy when distilled from the teacher model trained by different $\beta$ values.

We study the effect of $\beta$ here. For all experiments, we randomly sampled 50 different $\alpha$,As we can see, the knockoff student accuracy drops drastically when $\beta \geq 1$.

## L.2 NUMBER OF POWER SAMPLES $\alpha$

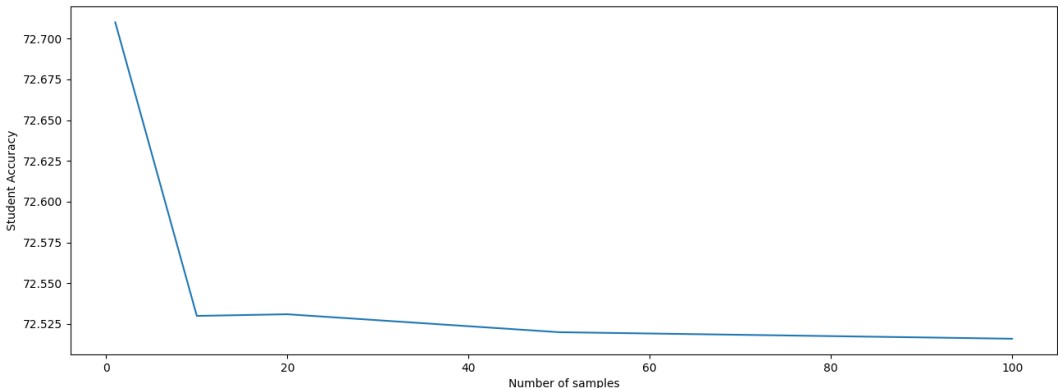

Figure 5: The student accuracy when distilled from the teacher model trained by different number of sample $\alpha$.

Here, we examine the effect of the number of samples on student performance. For a single sample, we use the conventional CE loss to train the teacher model. The student accuracy decreases monotonically as the number of samples increases.

