# OpenReview forum: "Towards Undistillable Models by Minimizing Conditional Mutual Information"
_ICLR.cc/2025/Conference — ICLR 2025 Conference Withdrawn Submission_

### Official Review · Reviewer_nSpi · 2024-10-30

**Soundness:** 3
**Presentation:** 2
**Contribution:** 3
**Rating:** 6
**Confidence:** 3

**Summary:**

The paper presents a method for protecting black-box models against knowledge distillation by student models. The authors define a DNN as distillable if a student model learned from its output outperforms the same model learned from ground truth labels. The proposed objective for the teacher model consists of a standard CE loss and a regularization term based on a tempered conditional mutual information (CMI) between the input and network predictions. Its aim is to make the output of the network close to one-hot encoding. Since the proposed objective is not tractable, through a series of approximations the authors propose a tractable objective. The authors demonstrate their method on CIFAR-100, TinyImageNet, and ImageNet using various teacher and student models, and against other baselines methods.

**Strengths:**

* The paper introduces a novel objective based on conditional mutual information that includes optimizing over a power-transformed probability distribution.
* Approximating the intractable terms of the objective is original although I am not sure that it is justified. I validated Theorem 4.1.
* I am not an expert in this field, but the experimental part seems very comprehensive in terms of datasets, student and teacher networks, defense strategies, and compared methods.
* The proposed approach seem to be the only one that makes the network not distillable on all benchmarks.

**Weaknesses:**

* I believe that further justification, evidence, or analysis (theoretical or empirical) is required to relate the approximation of the second term in the objective to the original one (as $\omega$ was taken to be a finite number). There is some discrepancy that needs to be settled as eventually instead of maximizing over $\mathbf{\alpha}$ (which makes sense), averaging is done over multiple values. Also, can you please share what values of $\omega$ were used in the paper? I didn't find this information.

* Experimental section:
  * I find the improvements over competing methods, and in particular label smoothing, marginal in most cases. I acknowledge that label smoothing does not adhere to the requirement of a network being undistillable, but it gets quite close to it. I am not convinced whether the minor improvements over it really make a difference in practice. Perhaps additional analysis or experiments in other scenarios can demonstrate the practical significance of your method over label smoothing.
  * This may be a criticism in general for defense methods in this domain and not specifically for this paper. It seems that the evaluation is done under the assumption that the student model has access to the input only ($\mathbf{x}$). How likely is that setup? In my opinion, a more realistic setup is distilling a model based on a new dataset altogether. I believe that a comparison in this setting will be much more informative.

* The paper has some writing issues in my opinion:
  * Some of the sentences are too long which makes it hard to understand at first pass (e.g., the first sentence of the abstract and the sentence in lines 46-50).
  * Some sentences are not clear until properly explained in the paper. For instance, "cluster of its output probability distributions in response
to all sample instances" or "cluster corresponding to each label should ideally collapse into one probability distribution", both are in the abstract.
  * In several locations there seems to be confusion in the citation format or whether a citation is justified at all (e.g., lines 72, 80)
  * The link for the code doesn't work.
  * Table 1 is very busy. The authors should consider breaking it down into several tables/figures.

* The authors claim that their training method makes the network undistillable, but it is validated only empirically. No formal proof is given. This is not an actual weakness since I acknowledge that giving such proof is hard and perhaps even impossible. Hence, it would be beneficial to discuss the limitations of the work in general and the empirical validation specifically. Perhaps adding a section on potential failure cases or datasets/methods where CMIM might not hold would help to provide a more balanced perspective on the method's applicability.

**Questions:**

* The paper seems to rely heavily on Yang et al. (2023). What is the technical novelty of this paper besides including the power-transformed elements?

---

> ### Author Response · Authors · 2024-11-27
> **Reply to Reviewer nSpi (1/3)**
>
> Thank you for appreciating our work in that our proposed approach seem to be the only one that makes the network not distillable on all benchmarks. Also, thank you for your time reading our paper. Please find our responses to your concerns in the following.
>
> > ### Weakness 1: I believe that further justification, evidence, or analysis (theoretical or empirical) is required to relate the approximation of the second term in the objective to the original one (as $\omega$ was taken to be a finite number). There is some discrepancy that needs to be settled as eventually instead of maximizing over $\mathbf{\alpha}$ (which makes sense), averaging is done over multiple values. Also, can you please share what values of $\omega$ were used in the paper? I didn't find this information.
>
> **Ans:**
> Thank you for raising this important point. As stated in Theorem 4.1, when $ \omega \to \infty $, the second term in the objective function becomes equivalent to the RHS of Eq. (20). We kindly ask the reviewer to verify the derivation provided in the appendix, which outlines the steps leading to this approximation. For a sufficiently large value of $ \omega $, we achieve the approximation stated in Eq. (21). Additionally, the integral in Eq. (21) is approximated by the summation provided in Eq. (22).
>
> In our experiments, we set $ \omega = 25 $, which we found to be a sufficiently large value to effectively mimic the behavior of $ \omega \to \infty $. It is worth noting that setting $ \omega $ to very large numbers, such as 1000, can induce numerical issues (e.g., overflow) in Python programming, making it impractical for computation.
>
> To address the impact of $ \omega $ comprehensively, we have included an ablation study in the revised version of the paper's appendix (Appendix L). This study examines the effect of different values of $ \omega $ on the results, providing further evidence and justification for our choice. We hope this additional analysis clarifies our approach and resolves any concerns.
>
> > ### Weakness 2 (Experimental section, part 1): I find the improvements over competing methods, and in particular label smoothing, marginal in most cases. I acknowledge that label smoothing does not adhere to the requirement of a network being undistillable, but it gets quite close to it. I am not convinced whether the minor improvements over it really make a difference in practice. Perhaps additional analysis or experiments in other scenarios can demonstrate the practical significance of your method over label smoothing.
>
> **Ans:**  Thank you for your observation. We respectfully argue that the improvement over label smoothing (LS) is not marginal when viewed in the correct context. While it is true that for **some** KD methods, the accuracy of the knockoff student trained on DNNs protected by LS and CMIM might appear close, the critical distinction lies in the undistillability of the DNNs: no existing KD method can render a DNN trained by CMIM distillable, whereas DNNs trained using LS can be rendered distillable by certain KD methods. In other words, there exists at least one KD method that makes LS trained DNNs distillable.
>
> > ### Weakness 2 (Experimental section, part 2): This may be a criticism in general for defense methods in this domain and not specifically for this paper. It seems that the evaluation is done under the assumption that the student model has access to the input only ($\mathbf{x}$). How likely is that setup? In my opinion, a more realistic setup is distilling a model based on a new dataset altogether. I believe that a comparison in this setting will be much more informative.
>
> **Ans:**  Thank you for your insightful comment. We agree that exploring the distillation process when the student model is trained on a dataset different from the one used to train the teacher model would be an interesting and more realistic scenario. However, it is not immediately clear whether conventional KD methods would be as effective in such cases, as the knowledge transfer process might be hindered by the domain gap between the two datasets.
>
> The investigation of how KD operates under these conditions could indeed open a new avenue of research and expand the current understanding of defense methods in this domain. We appreciate your suggestion and recognize it as a potential direction for future work.
>
> > ### Weakness 3 (Writing, part 1):  Some of the sentences are too long which makes it hard to understand at first pass (e.g., the first sentence of the abstract and the sentence in lines 46-50).
>
> **Ans:** Thank you for pointing this out. We have revised and restructured the sentences mentioned, including the first sentence of the abstract and the one in lines 46–50, by breaking them into smaller, more concise sentences.

---

> > ### Author Response · Authors · 2024-11-27
> > **Reply to Reviewer nSpi (2/3)**
> >
> > > ### Weakness 3 (Writing, part 2):
> >
> > **Ans:** Thank you for your feedback. The concept of clusters in the output probability space of DNNs is well-established in the literature, where the response of DNNs to samples from different classes naturally forms clusters, each corresponding to a different class [1,2,3]. The same terminology, "cluster", is commonly used in these references and aligns with our usage.
> >
> > Therefore, while we acknowledge the need for clear communication, we believe the sentences in the abstract are consistent with the established terminology in the field.
> >
> > > ### Weakness 3 (Writing, part 3):  In several locations there seems to be confusion in the citation format or whether a citation is justified at all (e.g., lines 72, 80)
> >
> > **Ans:** Thank you for pointing this out. We have reviewed and corrected the citation format throughout the paper to ensure consistency and accuracy in the revised version. We appreciate your attention to detail and for bringing this to our attention.
> >
> > > ### Weakness 3 (Writing, part 4): The link for the code doesn't work.
> >
> > **Ans:** Thank you for noting this issue. Immediately after submission, we realized that the original link breached the anonymity requirement of the paper, so we promptly removed it. In the revised version, we have included a new link that adheres to the anonymity guidelines. We apologize for the oversight and appreciate your understanding.
> >
> > > ### Weakness 3 (Writing, part 5): Table 1 is very busy. The authors should consider breaking it down into several tables/figures.
> >
> > **Ans:** Thank you for your suggestion. While we acknowledge that Table 1 is dense, it has been carefully structured to ensure readability. The table is already partitioned into three sections, each corresponding to a different dataset, which helps organize the information clearly.
> >
> > Further partitioning, such as dividing it by KD methods (columns) or defense methods (rows), would not necessarily simplify the presentation. On the contrary, it might fragment the data and make cross-referencing between methods more difficult, potentially leading to confusion. We believe the current structure strikes a balance between comprehensiveness and readability.
> >
> > > ### Weakness 4: The authors claim that their training method makes the network undistillable, but it is validated only empirically. No formal proof is given. This is not an actual weakness since I acknowledge that giving such proof is hard and perhaps even impossible. Hence, it would be beneficial to discuss the limitations of the work in general and the empirical validation specifically. Perhaps adding a section on potential failure cases or datasets/methods where CMIM might not hold would help to provide a more balanced perspective on the method's applicability.
> >
> > **Ans:** Thank you for this valuable suggestion. We acknowledge that providing a formal proof of undistillability is extremely challenging, and our claim is validated solely through empirical evidence. While we believe our comprehensive experiments demonstrate the robustness of CMIM across a wide range of KD methods and datasets, we agree that discussing potential limitations would add balance and clarity to the paper.
> >
> > In the revised version of the paper, we have added the limitations of our work in the conclusion section which includes acknowledging the reliance on empirical validation, the absence of a formal theoretical proof, and the possibility that future KD methods or specific datasets might expose vulnerabilities in CMIM.
> >
> > We believe this addition provides a more balanced perspective on the method's applicability and encourages further exploration in this area. Thank you for highlighting this important point.

---

> > > ### Author Response · Authors · 2024-11-27
> > > **Reply to Reviewer nSpi (3/3)**
> > >
> > > > ### Questions:  The paper seems to rely heavily on Yang et al. (2023). What is the technical novelty of this paper besides including the power-transformed elements?
> > >
> > > **Ans:**
> > > Thank you for your question. While our work builds upon the framework introduced by [3], the concept of CMI deployed in our paper is fundamentally different. Specifically, the CMI formula used in [3] to calculate $ I(X, \hat{Y} \mid Y) $ is based on the assumption that $ X, \hat{Y}, Y $ form a Markov chain as $ Y \to X \to \hat{Y} $.
> > >
> > > In contrast, our work introduces the concept of power-transformed clusters, where we calculate the concentration of these clusters using $ I(X, \hat{Y}^{\alpha[Y]} \mid Y) $. Here, $ \hat{Y}^{\alpha[Y]} $ explicitly depends on $ Y $, and as a result, the Markov chain assumption no longer holds. This introduces significant challenges in the calculation of $ I(X, \hat{Y}^{\alpha[Y]} \mid Y) $, which are not addressed in previous works.
> > >
> > > To overcome these challenges, we propose a novel method for calculating $ I(X, \hat{Y}^{\alpha[Y]} \mid Y) $, as detailed in Section 4.1. This methodological innovation represents a key technical contribution of our paper, setting it apart from prior work and extending the applicability of CMI to non-Markovian settings.
> > >
> > >
> > > ------
> > > **References:**
> > >
> > > [1] Tsekouras, G. E., \& Tsimikas, J. (2013). On training RBF neural networks using input–output fuzzy clustering and particle swarm optimization. Fuzzy Sets and Systems, 221, 65-89.
> > >
> > > [2] Tao, S. (2019). Deep neural network ensembles. In Machine Learning, Optimization, and Data Science: 5th International Conference, LOD 2019, Siena, Italy, September 10–13, 2019, Proceedings 5 (pp. 1-12). Springer International Publishing.
> > >
> > > [3] Yang, E. H., Hamidi, S. M., Ye, L., Tan, R., \& Yang, B. (2023). Conditional Mutual Information Constrained Deep Learning for Classification. arXiv preprint arXiv:2309.09123.

---

> > > > ### Comment · Reviewer_nSpi · 2024-11-28
> > > >
> > > > I thank the authors for the answers. Perhaps I am missing something, but in the current version of the paper the code link still leads to a page with a 404 error. Also, I didn't find in Appendix L the ablation over $\omega$ (only over $\alpha$ and $\beta$).

---

> > > > > ### Author Response · Authors · 2024-11-28
> > > > > **Authors' Response to Comment by Reviewer nSpi**
> > > > >
> > > > > Thank you for taking the time to review our response and providing additional feedback.
> > > > >
> > > > > 1. **Code Repository Link:** We sincerely apologize for the oversight in the uploaded PDF, which included an outdated link to our repository. Please find the correct link below:
> > > > >
> > > > >    [https://anonymous.4open.science/r/CMIM-CBCA](https://anonymous.4open.science/r/CMIM-CBCA)
> > > > >
> > > > >    We regret any inconvenience or confusion this may have caused. We will update the repo for the Camera-ready version of the paper.
> > > > >
> > > > > 2. **Ablation Study on Parameter $\omega$:** Thank you for pointing out the missing ablation study on $\omega$. We have conducted a detailed analysis of $\omega$ using VGG-16 and SNV2 as the teacher-student pair on the CIFAR-100 dataset.
> > > > >
> > > > >    In this experiment, we examined the impact of the power coefficient $\omega$ on the knockoff student's performance, keeping $\beta = 2$ and $N = 25$ fixed while varying $\omega$. The results are summarized in the table below:
> > > > >
> > > > >    | Value of $\omega$    | 1     | 2     | 5     | 10    | 15    | 20    | 25    | 30    | 40   | 50   | 100  | 200  |
> > > > >    |-------------------------|-------|-------|-------|-------|-------|-------|-------|-------|-------|-------|-------|-------|
> > > > >    | Knock-off Student Accuracy | 72.65 | 72.67 | 72.55 | 72.56 | 72.55 | 72.52 | 72.53 | 72.52 | NaN   | NaN   | NaN   | NaN   |
> > > > >
> > > > >    **Observations:**
> > > > >    - When $\omega > 30$, the simulations often result in NaN values due to excessively large exponent values.
> > > > >    - At $\omega = 25$, the knockoff student's accuracy reaches its minimum, effectively approximating the behavior observed when $\omega = \infty$.
> > > > >
> > > > > We hope this addresses your concerns.
> > > > >
> > > > > We would also like to reiterate the significance of the contribution of the paper again. To the best of our knowledge, our method is the **only one** in the literature capable of training undistillable DNNs that remain robust against a wide range of knowledge distillation (KD) methods. This represents a major advancement in the field.
> > > > >
> > > > > If you have any further questions or require additional clarification, please do not hesitate to let us know. Lastly, **we kindly ask you to consider raising your score to reflect the impact of this contribution.**

---

### Official Review · Reviewer_aDtW · 2024-11-03

**Soundness:** 3
**Presentation:** 3
**Contribution:** 3
**Rating:** 6
**Confidence:** 3

**Summary:**

The authors tackle the topic of *defending* trained models from getting *stolen* through knowledge distillation. They investigate when the teacher models are *undistillable* by knowledge distillation and introduce the CMIM method to train teachers to concentrate the predicted probability vectors in close clusters to minimize the information available for distillation. They theoretically introduce and support their method and empirically test the procedure as well as other *defence* techniques against multiple *attacking* techniques.

**Strengths:**

- The topic of undistillable models is highly relevant, particularly given the growing online prevalence of large closed-source models.
- The paper is mostly well-written with mostly appropriately supported claims.
- The authors provide a nice balance between theoretical and empirical results.

**Weaknesses:**

- L040: Missing reference to theoretical paper by Borup and Andersen (2021), “Even your Teacher Needs Guidance: Ground-Truth Targets Dampen Regularization Imposed by Self-Distillation,” NeurIPS.
- *"An insight is provided that in order for a DNN to be undistillable, it is desirable for the DNN to possess the trait that each cluster of the DNN’s output probability distributions corresponding to each label is highly concentrated to the extent that all probability distributions within the cluster more or less collapse into one probability distribution close to the one-hot probability vector of that label.”* Although, I am unable to provide a reference, that a concentrated probability distribution is uninformative, and thus desirable to avoid distillation is, to the best of my knowledge, common knowledge in the field, and should not be considered a contribution of this paper.
- Replacing $\hat{Y}$ with $\hat{Y}^\alpha$ in the MI and CMI is simply replacing a variable in a function; the change itself is not inherently innovative and warrant a notion of "contribution". However, the implications and importance of doing so may hold some importance.
- Section 4.1 appears redundant, as the extension follows naturally by substituting variables (see also comment above).
- Variance estimates in Appendix J reveal that multiple cases deemed "undistillable" in Table 1 do not definitively qualify as such.
- Table 2 mistakenly labels "RSP" as "RSG" and "RSD."
- Omitting CE from Table 1 limits insight into the distillability of the standard training procedure, and it is unclear how much better the proposed method is to the simplest baseline.
- (Minor) When introducing notation, some notation is used before it is introduced. Consider reordering this section so that no notation convention is used before it has been introduced.
- Equation (3): avoid using $\times$ if it solely represents normal multiplication.
- L403: "intensive" -> "extensive."
- L520-L524 could be rephrased, as the current phrasing appear redundant and confusing.

**Questions:**

- Please elaborate on the computational requirements of doing the proposed alternating optimization, where the optimization of the Q’s is done over multiple minibatches (I assume for each alternating step)?
- Proving a negative result is challenging empirically; the inability to surpass LS with tested KD methods does not necessarily imply that no viable methods exist or that the models trained were optimally configured. Without theoretical proof of undistillability, the results can unfortunately only be seen as the current state, as new (or already existing and untested) methods for KD might render the claims of this paper invalid soon. Please elaborate on why the results should be considered sufficient to prove a negative result.
- Table 1: A concerning number of results are reported as less than 10, which is either incorrect/faulty reporting or potential issues with collapsed training. If collapsed training runs, this supports the concern above (about negative results), and the authors should investigate and elaborate clearly on this.
- What happens if $\alpha = 1$? Setting $\alpha > 1$ naturally forces the simplex to be more concentrated in the corners, but this post-transform is not applicable to the probabilities a knockoff-student would train on.

---

> ### Author Response · Authors · 2024-11-27
> **Reply to Reviewer aDtW (1/3)**
>
> We thank the reviewer for their valuable feedback and for acknowledging the relevance of our work on undistillable models in the context of the increasing prevalence of large closed-source models. We also appreciate the recognition of the quality of our writing and the balance we have achieved between theoretical and empirical results. Please find our responses to your concerns in the following.
>
> > ### Weakness 1: L040: Missing reference to theoretical paper by Borup and Andersen (2021), “Even your Teacher Needs Guidance: Ground-Truth Targets Dampen Regularization Imposed by Self-Distillation,” NeurIPS.
>
> **Ans:** Thank you for pointing this out. We have included the missing reference to the paper. We appreciate your attention to detail.
>
> > ### Weakness 2: "An insight is provided that in order for a DNN to be undistillable, it is desirable for the DNN to possess the trait that each cluster of the DNN’s output probability distributions corresponding to each label is highly concentrated to the extent that all probability distributions within the cluster more or less collapse into one probability distribution close to the one-hot probability vector of that label.” Although, I am unable to provide a reference, that a concentrated probability distribution is uninformative, and thus desirable to avoid distillation is, to the best of my knowledge, common knowledge in the field, and should not be considered a contribution of this paper.
>
> **Ans:** Thank you for your comment. It appears there is a misunderstanding regarding the concept discussed in our paper. Our approach focuses on making the **clusters** of output probabilities corresponding to each class more concentrated. This is fundamentally different from the notion of a "concentrated probability distribution", which we have not employed or referenced in our work.
>
> If you are aware of any method in the literature that explicitly addresses the concentration of output probability clusters, we would appreciate it if you could point it out. To the best of our knowledge, [1] is the first paper to introduce and formalize this concept, making it a key novelty of our work. We hope this clarification resolves any confusion regarding the contributions of our method.
>
> > ### Weakness 3: Replacing $Y$ with $\hat{Y}$ in the MI and CMI is simply replacing a variable in a function; the change itself is not inherently innovative and warrant a notion of "contribution". However, the implications and importance of doing so may hold some importance.
>
> **Ans:**
> We would like to clarify that the innovation of this paper does not stem from simply "replacing $ Y $ with $ \hat{Y} $" in the MI and CMI formulations. Rather, our key contribution lies in the novel approach of measuring the concentration of output clusters for different values of the power transform $ \alpha[Y] $. To achieve this, we introduce the concept of $ \mathrm{I}(X; \hat{Y}^{\alpha[Y]} \mid Y) $, which quantifies this concentration in a meaningful way.
>
> Furthermore, calculating $ \mathrm{I}(X; \hat{Y}^{\alpha[Y]} \mid Y) $ is fundamentally different from calculating $ \mathrm{I}(X; \hat{Y} \mid Y) $. Specifically, as shown in [1], the calculation of $ \mathrm{I}(X; \hat{Y} \mid Y) $ relies on the Markov chain relationship $ Y \to X \to \hat{Y} $. However, in our case, $ Y $, $ X $, and $ \hat{Y}^{\alpha[Y]} $ no longer form a Markov chain, as $ \alpha[Y] $ explicitly depends on $ Y $. This shift introduces significant challenges, as minimizing CMI under a non-Markovian setting has not been explored in the literature before. Addressing this challenge is a novel and important contribution of our paper, which we believe should be acknowledged.
>
> > ### Weakness 4: Section 4.1 appears redundant, as the extension follows naturally by substituting variables (see also comment above).
>
> **Ans:**  Thank you for your comment. We respectfully disagree with the assertion that Section 4.1 is redundant. As detailed in our response to the earlier point, the extension presented in Section 4.1 is not a straightforward substitution of variables. Instead, it addresses the significant conceptual and computational challenges introduced by the dependency of $ \alpha[Y] $ on $ Y $, which breaks the Markov chain assumption ($ Y \to X \to \hat{Y} $) typically used to calculate $ \mathrm{I}(X; \hat{Y} \mid Y) $.
>
> Section 4.1 provides a rigorous framework for calculating $ \mathrm{I}(X; \hat{Y}^{\alpha[Y]} \mid Y) $ under these non-Markovian conditions. This methodological innovation is critical to the proposed approach and addresses an unexplored area in the literature, as no prior work has dealt with minimizing the CMI in such settings. We hope this explanation underscores the necessity and value of Section 4.1 in the paper.

---

> > ### Author Response · Authors · 2024-11-27
> > **Reply to Reviewer aDtW (2/3)**
> >
> > > ### Weakness 5: Variance estimates in Appendix J reveal that multiple cases deemed "undistillable" in Table 1 do not definitively qualify as such.
> >
> > **Ans:** Thank you for highlighting this concern. Upon reviewing the variance estimates in Appendix J, we acknowledge that certain cases, such as the (RN50, VGG11) pair on CIFAR-100, might suggest that the CMIM-trained model could be rendered distillable under naive statistical interpretations. For example, when the DIST method is applied to the CMIM model, the accuracy is reported as $71.86 \pm 0.28$, which could potentially exceed the LS student accuracy of $71.94 \pm 0.09$ if variance is simply added to the mean.
> >
> > However, this approach of directly comparing mean values with added variances does not provide an accurate or fair assessment of undistillability. To address this, we conducted a more rigorous analysis where, across five different seeds, we compared the accuracy of the knock-off Student (VGG11 in this case) trained via label smoothing and the DIST method applied to RN50 trained with CMIM.
> >
> > The results of this comprehensive comparison are summarized in the following table, which demonstrates that the CMIM model achieves undistillability across all different seeds. We hope this detailed clarification and additional data alleviate the concerns raised regarding the robustness of our findings.
> >
> > |      | Seed 1 | Seed 2 | Seed 3 | Seed 4 | Seed 5 |
> > |------|--------|--------|--------|--------|--------|
> > | LS   | 72.28  | 71.83  | 72.06  | 72.25  | 71.53  |
> > | MCMI | 72.01  | 71.74  | 71.93  | 72.12  | 71.27  |
> >
> > > ### Weakness 6: Table 2 mistakenly labels "RSP" as "RSG" and "RSD."
> >
> > **Ans:** Thank you for catching this typo. We have corrected the labeling error in Table 2, replacing "RSG" and "RSD" with the correct term "RSP" in the revised version of the manuscript. We appreciate your attention to detail.
> >
> > > ### Weakness 7: Omitting CE from Table 1 limits insight into the distillability of the standard training procedure, and it is unclear how much better the proposed method is to the simplest baseline.
> >
> > **Ans:** Thank you for your observation. By definition, the baseline for evaluating distillability and undistillability should be the accuracy of the student model trained using the label smoothing (LS) method. This is because it is well-established in the literature that the accuracy obtained using the LS method typically exceeds that achieved using CE (at least this is the case for the models tested in our paper). Therefore, including CE results in Table 1 would not provide any additional valuable insight into the distillability comparison, as LS already represents a stronger and more relevant baseline. We hope this explanation clarifies our rationale.
> >
> > > ### Weakness 8: When introducing notation, some notation is used before it is introduced. Consider reordering this section so that no notation convention is used before it has been introduced.
> >
> > **Ans:**  Thank you for pointing this out. We have revised the relevant sections to ensure that all notation is introduced before it is used.
> >
> > > ### Weakness 9: Equation (3): avoid using $\times$ if it solely represents normal multiplication. \\L403: "intensive" $\to$ "extensive." \\L520-L524 could be rephrased, as the current phrasing appear redundant and confusing.
> >
> > **Ans:** Thank you for identifying these issues. We have addressed all of them in the revised version of the paper. Specifically:
> >
> > - Equation (3) has been clarified to avoid any confusion if it represents normal multiplication.
> >
> > - The word "intensive" on L403 has been corrected to "extensive."
> >
> > - Lines 520–524 have been rephrased to eliminate redundancy and improve clarity.
> >
> > We appreciate your careful review and feedback.
> >
> >
> > > ### Question 1:  Please elaborate on the computational requirements of doing the proposed alternating optimization, where the optimization of the Q’s is done over multiple minibatches (I assume for each alternating step)?
> >
> > **Ans:**
> > Thank you for your question. To address this, we have added a new section in the appendix (Appendix K) that provides a detailed explanation of the computational requirements for the proposed alternating optimization.
> >
> > In summary, CMIM increases the training time by approximately 20\%, as the optimization of \( Q \)'s involves additional computations over multiple minibatches during each alternating step. We believe this is a reasonable trade-off given the significant benefits of the method. We encourage you to refer to Appendix K for a more comprehensive discussion of these requirements.

---

> > > ### Author Response · Authors · 2024-11-27
> > > **Reply to Reviewer aDtW (3/3)**
> > >
> > > > ### Question 2: Proving a negative result is challenging empirically; the inability to surpass LS with tested KD methods does not necessarily imply that no viable methods exist or that the models trained were optimally configured. Without theoretical proof of undistillability, the results can unfortunately only be seen as the current state, as new (or already existing and untested) methods for KD might render the claims of this paper invalid soon. Please elaborate on why the results should be considered sufficient to prove a negative result.
> > >
> > > **Ans:** Thank you for raising this important point. Providing a theoretical proof of undistillability is a significant challenge and falls beyond the scope of this work. The primary aim of our paper is to demonstrate empirically that our approach achieves undistillability against all existing KD attack methods in the literature. Importantly, we also show that our method succeeds where all existing defense methods have failed.
> > >
> > > While we acknowledge that the possibility of future KD methods rendering our approach vulnerable cannot be entirely ruled out, the empirical evidence presented here demonstrates the robustness of our approach against the current state of the art. We believe this comprehensive evaluation across all known KD methods provides strong evidence of the effectiveness of our method, even if it does not constitute a formal proof of a negative result.
> > >
> > > > ### Question 3: Table 1: A concerning number of results are reported as less than 10, which is either incorrect/faulty reporting or potential issues with collapsed training. If collapsed training runs, this supports the concern above (about negative results), and the authors should investigate and elaborate clearly on this.
> > >
> > > **Ans:** We would like to clarify that the reported numbers in Table 1 are correct and have been thoroughly validated through extensive experiments. The results indicate that all methods in the literature, except CMIM (our proposed method), fail to successfully render the model undistillable.
> > >
> > > The low values simply illustrate that certain KD methods (e.g., method DKD) are unable to extract and utilize information from a teacher model defended by another method (e.g., method MAD or APGP when DKD is used as the underlying KD method).
> > >
> > > We hope this clarification resolves any concerns regarding the reported results.
> > >
> > > > ### Question 4: What happens if $\alpha = 1$? Setting $\alpha > 1$ naturally forces the simplex to be more concentrated in the corners, but this post-transform is not applicable to the probabilities a knockoff-student would train on.
> > >
> > > **Ans:** Thank you for this insightful question. As shown by [2], logit temperature scaling with temperature $ T $ is mathematically equivalent to applying a power transform to the output probability distribution with power $ \alpha = \frac{1}{T} $. This relationship is discussed in the introduction of our paper.
> > >
> > > When $ \alpha = 1 $, it corresponds to $ T = 1 $, meaning no temperature scaling is applied, and there is no additional smoothing or sharpening of the probability vectors. In this case, the output probabilities remain unchanged.
> > >
> > > As noted by the reviewer, when $ \alpha > 1 $ (equivalently $ T < 1 $), the power transform makes the output probabilities more  peaky, pushing them closer to the corners of the simplex.
> > >
> > > -------
> > > **References**
> > >
> > > [1] Yang, E. H., Hamidi, S. M., Ye, L., Tan, R., \& Yang, B. (2023). Conditional Mutual Information Constrained Deep Learning for Classification. arXiv preprint arXiv:2309.09123.
> > >
> > > [2] Zheng, K., \& Yang, E. H. (2024). Knowledge distillation based on transformed teacher matching. arXiv preprint arXiv:2402.11148.

---

### Official Review · Reviewer_UcAd · 2024-11-04

**Soundness:** 3
**Presentation:** 2
**Contribution:** 2
**Rating:** 5
**Confidence:** 4

**Summary:**

To protect the intellectual property (IP) of pretrained DNNs (teachers), the authors propose a method to prevent student models from using knowledge distillation (KD) to mimic the teacher models’ behavior. Specifically, they focus on a black-box scenario where the student model can only access the inputs and output logits of the teacher model. The proposed conditional mutual information minimization (CMIM) method constrains the output probability distributions of the teacher model so that each cluster associated with a label is highly concentrated (highly peaked around a single label). Intuitively, this eliminates the inter-class information from the teacher model's output logits such that the student model receives no more information than the labels themselves, thereby protecting the IP of the teacher model.

**Strengths:**

1. The idea is intuitive and with sufficient details.

2. The benchmark defense and knowledge distillation (attack) methods are exhaustive in the experiments.

3. The paper is well-organized and easy to follow.

**Weaknesses:**

1. The discussions of the proposed method’s limitations are missing. The proposed method collapses the logits so that each class’s output is highly concentrated (as shown in Fig.2), the teacher model might become overly confident in its predictions. This can lead to poor calibration and deteriorate generalization capability on out-of-distribution (OoD) data. Therefore, more settings and evaluations on the protected teacher model’s performance beyond prediction accuracy are necessary.

2. The proposed method involves multiple hyperparameters, e.g., such as the number of power samples $N$ and the range $[0,\beta]$, but the ablation studies on hyperparameter sensitivity are missing. For example, have you assessed how these hyperparameters impact the model’s undistillability and accuracy? If some parameters are particularly influential, could you highlight those findings?

3. The proposed method introduces computation overhead, but the comparisons of computational costs are missing. The method introduces extra computation for minimizing CMI and performing multiple transformations, what is the relative computational cost of training a CMIM-protected model compared to a standard model and other protection methods? As we can see, the experiments are primarily conducted on small datasets, with very limited testing on the larger ImageNet dataset.

**Questions:**

Please see the Weaknesses. Other questions:
The proposed method seems to be limited to a single-label classification setting. Can the method potentially extend to regression or multi-label classification, where outputs are continuous or to predict multiple classes simultaneously? Can the method be adapted to protect the IP of the state-of-the-art models, e.g., LLMs, CLIP, and Diffusion models, which require it most?

---

> ### Author Response · Authors · 2024-11-27
> **Reply to Reviewer UcAd (1/2)**
>
> We sincerely thank the reviewer for their valuable feedback. We also appreciate the positive comments regarding the intuitiveness of our idea, the thoroughness of our experiments, and the organization of the paper. Please find our responses to your comments in the following.
>
> > ### Weakness 1 (part a): The discussions of the proposed method’s limitations are missing.
>
> **Ans:**  We have revised the conclusion section of the paper to explicitly address some of the method's limitations. Thank you for bringing this into our attention.
>
> > ### Weakness 1 (part b): The proposed method collapses the logits so that each class’s output is highly concentrated (as shown in Fig.2), the teacher model might become overly confident in its predictions. This can lead to poor calibration and deteriorate generalization capability on out-of-distribution (OoD) data. Therefore, more settings and evaluations on the protected teacher model’s performance beyond prediction accuracy are necessary.
>
> **Ans:** We would like to clarify the distinction between "*highly concentrated output clusters*" and "*overly confident predictions*". A highly concentrated output cluster **does not necessarily** imply that the model produces overly confident predictions. This is because the clusters can be concentrated around points that are not close to one-hot labels (the corners of the probability simplex). As a result, the model can have concentrated outputs without being overly confident. These are two separate concepts.
>
> Our experiments support this distinction: the proposed method creates more concentrated output clusters, yet the accuracy of the model improves on the held-out test dataset compared to models trained with the conventional cross-entropy (CE) loss. This observation aligns with findings in [1], where it was demonstrated that training DNNs to produce highly concentrated output clusters can enhance their test accuracy.
>
> Regarding out-of-distribution (OoD) data, we acknowledge the importance of such evaluations; however, the primary focus of this work is on training undistillable DNNs. While extending the evaluation to OoD scenarios is a valuable future direction, it is beyond the current scope of this study.
>
> > ### Weakness 2: The proposed method involves multiple hyperparameters, e.g., such as the number of power samples $N$ and the range $[0,\beta]$, but the ablation studies on hyperparameter sensitivity are missing. For example, have you assessed how these hyperparameters impact the model’s undistillability and accuracy? If some parameters are particularly influential, could you highlight those findings?
>
> **Ans:** In response to your comment, we have conducted detailed ablation studies for the key hyperparameters, including the number of power samples $N$ and the range $[0, \beta]$, to assess their impact on the model’s undistillability and accuracy. The results of these studies have been included in the appendix (Appendix L) to provide a comprehensive understanding of the sensitivity and influence of these parameters. We hope this addition addresses your concern and strengthens the paper.
>
> > ### Weakness 3: The proposed method introduces computation overhead, but the comparisons of computational costs are missing. The method introduces extra computation for minimizing CMI and performing multiple transformations, what is the relative computational cost of training a CMIM-protected model compared to a standard model and other protection methods? As we can see, the experiments are primarily conducted on small datasets, with very limited testing on the larger ImageNet dataset.
>
> **Ans:**  We have added another section in the appendix (Appendix K) which discuss the computational overhead of CMIM. The experiment results is comprehensive and intensive. In summary, CMIM increases the training time by approximately 20\%.
>
> Additionally, we note that among the existing KD protection methods in the literature, only CMIM (our method) and ST are scalable to larger datasets like ImageNet. This scalability is due to the significant computational complexity of other benchmark methods, which limits their applicability to smaller datasets. We hope this clarification and the added discussion in the appendix address your concern.

---

> > ### Author Response · Authors · 2024-11-27
> > **Reply to Reviewer UcAd (2/2)**
> >
> > > ### Questions: The proposed method seems to be limited to a single-label classification setting. Can the method potentially extend to regression or multi-label classification, where outputs are continuous or to predict multiple classes simultaneously? Can the method be adapted to protect the IP of the state-of-the-art models, e.g., LLMs, CLIP, and Diffusion models, which require it most?
> >
> > **Ans:** Thank you for these thought-provoking questions. The concept of undistillable DNNs, as introduced in [1], is relatively new. The primary goal of this paper is to demonstrate that it is possible to train undistillable DNNs in conventional single-label classification tasks. Due to the scope of this work, it is not feasible to extend the methodology to a wide range of learning tasks within the same paper.
> >
> > Adapting the method to multi-label classification, regression, or protecting the intellectual property of state-of-the-art models such as LLMs, CLIP, and Diffusion models presents unique challenges and would require tailored approaches. These extensions are valuable directions for future work, and we are eager to explore them in subsequent research.
> >
> >
> > -------------
> > **References:**
> >
> > [1] Yang, E. H., \& Ye, L. (2024). Markov knowledge distillation: make nasty teachers trained by self-undermining knowledge distillation fully distillable. In European Conference on Computer Vision. Springer (Vol. 3).

---

> ### Author Response · Authors · 2024-12-03
>
> Dear Reviewer UcAd,
>
> Thank you again for reviewing our work and providing your valuable insights.
>
> With the rebuttal deadline fast approaching, we kindly urge you to share your feedback on our responses to your comments at your earliest convenience. If you have any additional concerns or questions, please let us know. We are committed to addressing all your points comprehensively and are happy to provide further clarifications or details as needed. Otherwise, we kindly ask you to consider raising the score if our responses have resolved your concerns.
>
> Thank you for your time and consideration.
>
> Best regards,
> The Authors

---

> > ### Author Response · Authors · 2024-12-03
> >
> > Dear Reviewer UcAd,
> >
> > Thank you again for your thoughtful review and valuable feedback on our work.
> >
> > In our response, we have carefully addressed all the concerns raised in your review. With the rebuttal deadline rapidly approaching, we kindly request your feedback on our responses at your earliest convenience. If you have any additional questions or concerns, please do not hesitate to let us know. We are fully committed to ensuring all your points are thoroughly addressed and are happy to provide further clarifications or details as needed.
> >
> > We greatly appreciate your time and consideration.
> >
> > Best regards,
> > The Authors

---

> ### Author Response · Authors · 2024-12-04
>
> Dear Reviewer’s UcAd,
>
> Thank you for your detailed and constructive feedback on our submission, which has significantly improved the quality of our work. We have carefully addressed all the concerns raised in your review and provided detailed responses to each point.
>
> If possible, we kindly request a re-evaluation of the score, considering that the issues highlighted have been thoroughly resolved.
> We truly appreciate your time and effort in reviewing our work and are grateful for your consideration.
>
> Best regards.

---

### Official Review · Reviewer_x3fp · 2024-11-05

**Soundness:** 2
**Presentation:** 1
**Contribution:** 2
**Rating:** 5
**Confidence:** 4

**Summary:**

This paper introduces a defense method against knowledge distillation (KD) attacks, where the goal is to avoid the undesired usage of the outputs of deep neural networks (DNNs) by making them undistillable. The authors propose a training method that aims to minimize the conditional mutual information (CMI) across all temperature-scaled clusters, resulting in a model that cannot be effectively distilled by existing KD methods. The CMIM model is shown to be undistillable through extensive experiments on CIFAR-100, TinyImageNet, and ImageNet datasets, while outperforming state-of-the-art methods and even improving upon the conventional cross-entropy (CE) loss in terms of prediction accuracy.

**Strengths:**

1. The paper's goal is to prevent the misuse of models and serves a partial privacy protection technique, which is significant for the reliable use of AI models.

2. The paper provides both theoretical and empirical evidence to demonstrate the benefits of the proposed method in enhancing the undistillability of models. T

**Weaknesses:**

1. The writing quality of the paper could be enhanced.

2. From a methodological perspective, the overall contribution of the paper is somewhat limited. The paper utilizes an existing metric, CMI, to measure the compactness of model outputs and aims to enhance model undistillability by maximizing this compactness metric. This approach appears too trivial and straightforward. It is not clear how this method fundamentally differs from directly employing a maximum entropy term or label smoothing technique to increase output concentration. Moreover, in the field of machine learning, particularly in computer vision, numerous loss functions have been studied to enhance model output compactness, such as the Large-Margin-Softmax-based methods.

3. The paper's finding that the teacher model trained with the proposed method achieves higher accuracy is not surprising. The mechanism by which the proposed method operates is similar to that of label smoothing, which is known to enhance accuracy. The authors might refer to the paper "When does label smoothing help" to understand that label smoothing can also produce the feature compression effect as shown in Figure 2 of this manuscript. The enhancement in accuracy due to the proposed method is expected and aligns with the effects of label smoothing, which is not a novel discovery in the field.

**Questions:**

Please refer to the weaknesses.

---

> ### Author Response · Authors · 2024-11-27
> **Reply to Reviewer x3fp (1/2)**
>
> We appreciate the reviewers recognizing the significance of our work in preventing the misuse of models and contributing to partial privacy protection. We also thank the reviewers for acknowledging the theoretical and empirical contributions of our work in demonstrating the effectiveness of our proposed method in enhancing the undistillability of models.
> Please find our responses to your comments in the sequel.
>
> >### Weakness 1: The writing quality of the paper could be enhanced.
>
> **Ans:** Thank you for pointing out the need for improving the writing quality. We have revised several lengthy and complex sentences throughout the paper to enhance clarity and make it more reader-friendly. Additionally, we will make further editorial refinements as needed following the paper’s acceptance.
>
> >### Weakness 2 (part a): From a methodological perspective, the overall contribution of the paper is somewhat limited. The paper utilizes an existing metric, CMI, to measure the compactness of model outputs and aims to enhance model undistillability by maximizing this compactness metric. This approach appears too trivial and straightforward.
>
> **Ans:** We appreciate the reviewer’s comments and would like to clarify the unique contributions of our work.
>
> First, our paper is the only one to propose a method for training undistillable DNNs capable of resisting a wide range of knowledge distillation (KD) methods. We invite the reviewer to identify any existing approaches in the literature that achieve this specific goal. Given the novelty and practical importance of our work, we firmly believe that our contributions should not be considered trivial.
>
> Second, while we leverage the concept of CMI from [1], the way it is calculated in our work significantly differs from how it is calculated [1]. In [1], the calculation of $ I(X, \hat{Y} \mid Y) $ is based on the assumption of a Markov chain $ Y \to X \to \hat{Y} $. However, in our work, we measure the compactness of clusters using $ I(X, \hat{Y}^{\alpha[Y]} \mid Y) $, where the term $ \hat{Y}^{\alpha[Y]} $ explicitly depends on $ Y $, breaking the Markov chain assumption. Consequently, calculating $ I(X, \hat{Y}^{\alpha[Y]} \mid Y) $ poses additional challenges. We address these challenges by proposing a novel method for calculating this metric, as detailed in Section 4.1.
>
> Finally, we introduce a novel training method called the CMI minimization method. This approach trains a DNN by jointly minimizing the cross-entropy loss and the CMI values of all power-transformed clusters. This joint minimization process is non-trivial and requires careful formulation, as discussed in the paper.
>
> These elements collectively establish the methodological contributions of our work and demonstrate its significance.
>
> >### Weakness 2 (part b): It is not clear how this method fundamentally differs from directly employing a maximum entropy term or label smoothing technique to increase output concentration. Moreover, in the field of machine learning, particularly in computer vision, numerous loss functions have been studied to enhance model output compactness, such as the Large-Margin-Softmax-based methods.
>
> **Ans:** We appreciate the reviewer’s comments and would like to clarify the distinction between our method and existing approaches.
>
> While there are methods in the literature that aim to enhance the intra-class compactness of DNN output clusters, this alone is \textbf{insufficient} to achieve undistillability. Our work is the first to propose a method specifically designed to train undistillable DNNs that remain robust against KD techniques.
>
> Regarding the methods mentioned by the reviewer:
>
> 1. Label Smoothing and Large-Margin-Softmax-based methods: These approaches may lead to more concentrated clusters. However, this is \textbf{inadequate} for ensuring compactness under power transformations, a key aspect required to achieve undistillability. As a result, models trained with these methods may still be vulnerable to KD techniques.
>
> 2. Maximum Entropy Term: Increasing the entropy of outputs merely reduces the model's confidence, making its predictions less certain. This does not directly relate to or contribute to the concentration of output clusters, which is the cornerstone of our approach for achieving undistillability.
>
> Our method fundamentally differs by addressing these limitations and incorporating a novel strategy to ensure compactness under power transformations, which is critical for training undistillable DNNs. We hope this clarification highlights the unique contributions of our work.

---

> > ### Author Response · Authors · 2024-11-27
> > **Reply to Reviewer x3fp (2/2)**
> >
> > >### Weakness 3: The paper's finding that the teacher model trained with the proposed method achieves higher accuracy is not surprising. The mechanism by which the proposed method operates is similar to that of label smoothing, which is known to enhance accuracy. The authors might refer to the paper "When does label smoothing help" to understand that label smoothing can also produce the feature compression effect as shown in Figure 2 of this manuscript. The enhancement in accuracy due to the proposed method is expected and aligns with the effects of label smoothing, which is not a novel discovery in the field.
> >
> > **Ans:** Thank you for the comment. We would like to clarify that the primary focus of this paper is not on increasing the accuracy of DNNs but on developing a method to train undistillable DNNs. While many existing methods in the literature can enhance a DNN’s accuracy, they do not address the critical challenge of making DNNs undistillable.
> >
> > Our approach is the first in the literature that effectively trains undistillable models robust against a wide range of existing KD methods. The improvement in accuracy observed in our results is a by-product of our method and not its primary goal. This improvement arises from the unique properties of our approach rather than replicating the effects of label smoothing or similar techniques.
> >
> > We believe this distinction highlights the novelty and importance of our contribution to the field.
> >
> >
> >
> > -------------------
> > **References:**
> >
> > [1] Yang, E. H., Hamidi, S. M., Ye, L., Tan, R., \& Yang, B. (2023). Conditional Mutual Information Constrained Deep Learning for Classification. arXiv preprint arXiv:2309.09123.

---

> ### Author Response · Authors · 2024-12-03
>
> Dear Reviewer x3fp,
>
> Thank you for taking the time to review our work. We have conducted additional experiments to provide further clarification on the points you raised.
>
> We would greatly appreciate your feedback on our rebuttal. If you have any further concerns or questions, please let us know. We are eager to ensure we have thoroughly addressed your comments and are happy to provide any additional clarifications. Otherwise, we kindly ask you to consider raising the score if our responses have resolved your concerns.
>
> Thank you for your consideration.
>
> Best regards,
> The Authors

---

> > ### Comment · Reviewer_x3fp · 2024-12-03
> >
> > Thanksfor your reply.
> >
> > I apologize if my previous statement caused any confusion. What I mentioned is that "Label smoothing is equivalent to adding a confidence penalty term to the original cross-entropy,". I did not equate label smoothing with maximizing entropy, because as you mentioned, the cross-entropy between the uniform distribution and the DNN output distribution is NOT equal to the entropy of the DNN output distribution. However, you cannot deny that the term which minimizes the cross-entropy between the uniform distribution and the DNN output distribution can be considered as a confidence penalty term.
> >
> > Furthermore, what I said was that "the author's statement that increasing the entropy of outputs does not directly relate to the concentration of output clusters is somewhat wrong." My point is that increasing the entropy of outputs is related to the concentration of output clusters (not equivalent, hence the example you provided at the end does not address my question), because techniques like label smoothing lead to a concentration of output clusters. Therefore, it is an oversimplification to say that "increasing the entropy of outputs does not directly relate to the concentration of output clusters".
> >
> > However, the author's experimental results partially address my concern regarding to the comparison with LS. The additional experiments show that the proposed method can indeed achieve a better trade-off compared to the simple LS.
> >
> > Based on the overall rebuttal, I would like to increase the score to 5, but I still believe that the paper can be significantly improved in the overall writing quality and experimental analysis, even though the phenomenon of undistillable models analyzed in this paper is somewhat interesting.

---

> > > ### Author Response · Authors · 2024-12-03
> > >
> > > Thank you for your detailed and constructive feedback. We appreciate the time you have taken to clarify your perspective and for recognizing the value of our work.
> > >
> > > 1. Regarding the relationship between label smoothing and entropy:
> > >    We acknowledge and agree with your clarification that the term minimizing the cross-entropy between a uniform distribution and the DNN output distribution can indeed be interpreted as a confidence penalty term. Our intention was to emphasize that label smoothing introduces a regularization effect, rather than equating it directly with maximizing entropy.
> > >
> > > 2. On the relationship between entropy and the concentration of output clusters:
> > >    We understand your concern and appreciate the nuanced point you are raising. While we agree that techniques like label smoothing can lead to more concentrated output clusters in practice, our statement was intended to highlight that increasing the entropy of the outputs does not *necessarily* lead to such concentration under all conditions. To address your comment, we will clarify this statement in the revised manuscript to reflect that there is a complex relationship between entropy and cluster concentration, influenced by the specific regularization technique employed.
> > >
> > > 3. Writing quality and experimental analysis:
> > >    We are grateful for your constructive criticism regarding the writing quality and the need for further experimental analysis. In response, we plan to carefully revise the manuscript to improve its overall clarity and coherence.
> > >
> > > 4. Rebuttal impact:
> > >    We are pleased to see that our additional experiments addressing the comparison with label smoothing have partially alleviated your concerns. We will include the extended analysis and discussion in the revised paper to provide a more comprehensive understanding of the trade-offs offered by our method.
> > >
> > > Thank you once again for your thoughtful review and for increasing your score. Your feedback has been invaluable in helping us identify areas for improvement and strengthening the presentation of our work.

---

> > > > ### Author Response · Authors · 2024-12-04
> > > >
> > > > Dear Reviewer x3fp,
> > > >
> > > > Thank you for your thoughtful feedback and for the opportunity to address the points raised in your review.
> > > >
> > > > As we discussed, through comprehensive experiments, we have demonstrated that CMIM is the only method in the literature which make the teacher model undistillable across a broad range of logit-based KD methods while simultaneously improving the teacher’s accuracy.
> > > >
> > > > While label smoothing can indeed enhance output cluster compactness to some extent under specific power transform factors (temperatures), it does not guarantee compactness across all possible factors. To further strengthen  this claim, we conducted additional experiments, which are detailed below.
> > > >
> > > >
> > > > 1. We tested label smoothing with a smaller smoothing factor and observed that, in some cases, label smoothing can increase the model's CMI value (decrease compactness). We also report the entropy of the output probability vectors.
> > > >
> > > > |          | CE    | LS ($\alpha=0.001$) | LS($\alpha=0.005$) | LS($\alpha=0.007$) |
> > > > |----------|-------|--------------------|-------------------|-------------------|
> > > > | CMI          | 0.071   | 0.071        | 0.076        | 0.070            |
> > > > | Entropy   | 0.0246 | 0.0373     | 0.0470      | 0.0692          |
> > > > | Accuracy | 77.81   | 77.81       | 77.82        | 77.83             |
> > > >
> > > > 2. We applied MKD to distill a teacher trained with a label smoothing factor of 0.5. Once again, we found the LS-trained teacher to be distillable.
> > > >
> > > > |ResNet50 (LS 0.5)| LS    | MKD |
> > > > |----------|-------|--------------------|
> > > > | VGG11      | 71.94 | 72.04     |
> > > >
> > > > We kindly request your consideration for an increased score if all your concerns have now been fully addressed.

---

### Note · Authors · 2025-01-29

I have read and agree with the venue's withdrawal policy on behalf of myself and my co-authors.